# Toward bilipshiz geometric models

## Abstract

Many neural networks for point clouds are, by design, invariant to the symmetries of this datatype: permutations and rigid motions. The purpose of this paper is to examine whether such networks preserve natural symmetry aware distances on the point cloud spaces, through the notion of bi-Lipschitz equivalence. This inquiry is motivated by recent work in the Equivariant learning literature which highlights the advantages of bi-Lipschitz models in other scenarios.

We consider two symmetry aware metrics on point clouds: (a) The Procrustes Matching (PM) metric and (b) Hard Gromov Wasserstien distances. We show that these two distances themselves are not bi-Lipschitz equivalent, and as a corollary deduce that popular invariant networks for point clouds are not bi-Lipschitz with respect to the PM metric. We then show how these networks can be modified so that they do obtain bi-Lipschitz guarantees. Finally, we provide initial experiments showing the advantage of the proposed bi-Lipschitz model over standard invariant models, for the tasks of finding correspondences between 3D point clouds.

## 1 Introduction

We consider neural networks defined on points sets- a collections of $n$ points in $\mathbb{R}^d$ represented by a matrix $\boldsymbol{X} \in \mathbb{R}^{n \times d}$, which are invariant or equivariant to permutation, rotation and translation of the points. These models are suitable for a large body of invariant and equivariant tasks, such as 3D point cloud processing in computer vision, learning of particle dynamics, and chemoinformatics, and as a result there is a large body of work targeting point sets with this type of symmetries, notable examples including EGNN Satorras et al. (2021), Vector Neurons Deng et al. (2021), MACE Batatia et al. (2022), DimeNet Gasteiger et al. (2020b;a) and SphereNet Liu et al. (2021).

To navigate the large space of possible invariant neural networks and understand their theoretical expressive power, the notion of *completeness* was introduced. An invariant model on point sets will be complete, if it can assign distinct outputs to any given pair of points sets which are not related by a group transformation. The importance of this inquiry can be understood via the observation that if $f_\theta$ is a model which is not complete, and it assigns the same value to two different point clouds, then it will struggle to approximate functions which assign different values to these point clouds. Conversely, models which are complete can be used to approximate all invariant functions Chen et al. (2019); Dym & Gortler (2024).

In the last few years the machine learning community has developed a reasonably good picture of completeness of different invariant models. Models like Tensor Field Networks Thomas et al.

(2018) and Gemnet Gasteiger et al. (2021) were shown to be complete Dym & Maron (2020), while others were shown to be incomplete Pozdnyakov & Ceriotti (2022); Li et al. (2023). Recently, there have been several works showing completeness of invariant models which apply $k$-GNN to rotation invariant weighted graphs induced by $d$ dimensional point clouds, providing that $k + 1 \geq d$ Hordan et al. (2023); Delle Rose et al. (2023); Li et al. (2023). Similar ideas were used in Li et al. to prove completeness via subgraph GNNs, and compeleteness of Spherenet Liu et al. (2021) and Dimenet Gasteiger et al. (2020b). However, completeness guarantees do not address the quality of separation: namely, completeness implies that if $X, X'$ are point sets which are not equivalent, then $f_\theta(X) \neq f_\theta(X')$. However, it does not guarantee that if $X, X'$ are 'close' (or 'far') then $f_\theta(X)$ and $f_\theta(X')$ will be 'close' ( or 'far'). To ensure preservation of distances under the map $f_\theta$, we would like to guarantee that it is bi-Lipschitz in an appropriate sense. Bi-Lipschitz models can provide provable guarantees for metric based learning Cahill et al. (2024), and recent work has proven the value of bi-Lipschitz models for permutation invariant learning for multisets and graphs Davidson & Dym (2024); Sverdlov et al. (2024); Amir & Dym. Motivated by these works, our goal in this paper is to understand how to construct point set networks which are not only complete, but are also bi-Lipschitz.

## 1.1 Main Results

The notion of bi-Lipchitzness requires a 'natural' metric defined on the space of points sets, up to permutation and rigid motion symmetries. In this work, we consider two such metrics, often discussed in the literature: (a) The Procrustes Matching (PM) metric, obtained by simple quotienting a norm on point set space by the group of permutations and rigid motions, and (b) the Hard-Gromov-Wasserstein distance, where Euclidean distances, which are inherently invariant to rigid motions, are compared up to permutations. Our first result is to show that these metrics are not bi-Lipschitz equivalent, and find the exact Hölder exponents (2 and 1) which govern the relationship between these metrics. Accordingly, an invariant model for point sets cannot be bi-Lipschitz with respect to both of these metric simultaneously. We choose in this paper to focus on bi-Lipschitzness with resect to the Procrustes Matching metric, as we believe this metric is more natural for point sets, while Gromov-Wasserstein distances are more suitable for comparison of general metric spaces.

Next, we consider a popular class of invariant models for point sets, which employs message passing neural networks to the distance matrix induced by the point sets. Such models are known to be complete when $d = 2$ (but not for larger $d$). We show that in the planar case, these complete models are not bi-Lipschitz with respect to the PM metric. We then show how to modify these models, in the spirit of the work of Hordan et al. (2023), to obtain a bi-Lipschitz model in the planar case. We will then show how to generalize the bi-Lipschitz construction to the case $d \geq 3$ (under some additional assumptions). Finally, we present some preliminary experiments on the task of learning the rotation-permutation correspondence between pairs of point clouds. We show that in this task the proposed bi-Lipschitz model outperforms a standard invariant model which are not bi-Lipschitz.

## 1.2 Related work

Motivated by invariant learning tasks, there have been several recent works discussing bi-Lipschitzness for invariant models Cahill et al. (2024); Dym et al. (2025a); Blum-Smith et al. (2025). In particular, several papers discussed *permutation invariant* models for both sets Balan et al. (2022); Amir & Dym; Cahill et al. (2024) and graphs Davidson & Dym (2024); Sverdlov et al. (2024). Bi-Lipschitz

invariants with respect to the action of *rotations* were discussed in Derksen (2024); Amir et al. (2025). In this paper, our goal is to find bi-Lipschitz invariant models with respect to the joined action of permutations, rotations and translations. Though upper-Lipschitz models were discussed in this setting Widdowson & Kurlin (2023), to the best of our knowledge this paper is the first to address the bi-Lipschitz problem for the joint symmetry group of permutations, rotations and translations.

## 1.3  Definitions

We now introduce some definitions necessary for formally stating and proving our results.

For given natural $n, d$, A point set is a matrix $\mathbf{X} \in \mathcal{V} = \mathbb{R}^{d \times n}$, whose $n$ columns, denoted by $x_1, \ldots, x_n$ represent a collection of $n$ points in $\mathbb{R}^d$. We are interested in invariance with respect to

1. **Permutation:** The application of a permutation $\tau \in S_n$ to $\mathbf{X}$, denoted by

$$\tau(x_1, \ldots, x_n) = (x_{\tau(1)}, \ldots, x_{\tau(n)}).$$

2. **Rotation:** The application of the same (improper) rotation $R \in O(d)$ to each point, namely

$$R(x_1, \ldots, x_n) = (Rx_1, \ldots, Rx_n)$$

3. **Translation:** The application of the same translation $t \in \mathbb{R}^d$ to each point, namely

$$t(x_1, \ldots, x_n) = (x_1 + t, \ldots, x_n + t).$$

We denote the group generated by permutations, rotations and translations by $\mathcal{G}_\pm = \mathcal{O}(d) \rtimes \mathbb{R}^d \times S_n$, where the semi-product $\mathcal{O}(d) \rtimes \mathbb{R}^d$ is known as the 'rigid motions group'. We will also discuss a close variant of this setting, where we only allow proper rotations, namely matrices $R \in SO(d)$ which have determinant 1. We denote the slightly smaller group obtained by adding this restriction to proper rotations by $\mathcal{G}_+ = \mathcal{SO}(d) \rtimes \mathbb{R}^d \times S_n$.

In the following definitions, $\mathbb{G}$ will be a group acting on $\mathcal{V} = \mathbb{R}^{d \times n}$.

**Definition 1.1** (Isomorphic elements). For $\boldsymbol{X}, \boldsymbol{Y} \in \mathcal{V}$, we say that $\boldsymbol{X}, \boldsymbol{Y}$ are $\mathbb{G}$-isomorphic, and denote $\boldsymbol{X} \cong \boldsymbol{Y}$, if $\boldsymbol{X} = g\boldsymbol{Y}$ for some $g \in \mathbb{G}$.

**Definition 1.2** (Invariant metric). We say $d : \mathcal{V} \times \mathcal{V} \to \mathbb{R}^{\geq 0}$ is a $\mathbb{G}$ invariant metric if for all $\boldsymbol{X}, \boldsymbol{Y}, \boldsymbol{Z} \in \mathcal{V}$,

$$d(\boldsymbol{X}, \boldsymbol{Y}) = d(\boldsymbol{Y}, \boldsymbol{X})$$
$$d(\boldsymbol{X}, \boldsymbol{Y}) \leq d(\boldsymbol{X}, \boldsymbol{Z}) + d(\boldsymbol{Z}, \boldsymbol{Y})$$
$$d(\boldsymbol{X}, \boldsymbol{Y}) = 0 \iff \boldsymbol{X} \cong \boldsymbol{Y}$$

Equivalently, a $\mathbb{G}$ invariant metric is a metric on the quotient space $\mathcal{V}/\mathbb{G}$.

In this paper, we will consider two different $\mathcal{G}_\pm$ invariant metrics, and show that they are bi-Hölder equivalent but not bi-Lipschitz equivalent. We now define these notions formally.

**Definition 1.3** (Hölder metrics). Let $d_1, d_2$ be two non-negative functions on $\mathcal{V}$. We say that $d_1$ is $(\alpha_1, \alpha_2)$ bi-Hölder with respect to $d_2$ if there exists constants $C_1, C_2 > 0$ such that

$$C_1 \cdot d_1(\boldsymbol{X}, \boldsymbol{Y})^{\alpha_1} \leq d_2(\boldsymbol{X}, \boldsymbol{Y}) \leq C_2 \cdot d_1(\boldsymbol{X}, \boldsymbol{Y})^{\alpha_2}$$

We say $d_1, d_2$ are bi-Lipschitz equivalent if $d_1$ is bi-Hölder with respect to $d_2$ with $(\alpha_1, \alpha_2) = (1, 1)$.

Once we discuss invariant metrics, the next step will be to discuss the stability of invariant models with respect to these invariant metrics. An invariant function, and a complete invariant function, are defined as

**Definition 1.4** (Invariant and complete functions). We say a function $f : \mathcal{V} \to \mathbb{R}^m$ is $\mathbb{G}$ invariant, if

$$\forall \boldsymbol{X} \cong \boldsymbol{Y}, f(\boldsymbol{X}) = f(\boldsymbol{Y}).$$

We say that an invariant function is complete if the converse is also true: namely,

$$\forall \boldsymbol{X}, \boldsymbol{Y} \in \mathcal{V}, f(\boldsymbol{X}) = f(\boldsymbol{Y}) \iff \boldsymbol{X} \cong \boldsymbol{Y}.$$

The stability of a invariant function $f$ is defined via the notions of upper and lower Lipschitz stability

**Definition 1.5** (Upper and lower lipshiz). Given some $\mathbb{G}$ invariant metric $d$, we say a $\mathbb{G}$ invariant function $f$ is upper-Lipschitz with respect to $d$, if there exists $C > 0$ such that

$$\|f(\boldsymbol{X}) - f(\boldsymbol{Y})\|_2 \leq C \cdot d(\boldsymbol{X}, \boldsymbol{Y}).$$

We say $f$ is lower lipshiz if there exists some $c > 0$ such that

$$c \cdot d(\boldsymbol{X}, \boldsymbol{Y}) \leq \|f(\boldsymbol{X}) - f(\boldsymbol{Y})\|_2$$

We say $f$ is bi-Lipschitz with repsect to $d$ if $f$ is both upper and lower Lipschitz.

We note that a Lower Lipshitz function $f$ must be complete, but not vice-versa. We also note that $f$ is bi-Lipschitz with respect to $d$ if and only if the invariant metric $d_f(\boldsymbol{X}, \boldsymbol{Y}) = \|f(\boldsymbol{X}) - f(\boldsymbol{Y})\|_2$ induced by $f$ is bi-Lipschitz equivalent to $d$ in the sense of Definition 1.3.

## 2 Geometric metrics

In this section we introduce two $\mathcal{G}_\pm$ invariant metrics, and discuss the Hölder relationships between them. We then discuss a related $\mathcal{G}_+$ invariant metric, and the concept of centralization.

**The PM metric** Arguably, the most popular $\mathcal{G}_\pm$ invariant metric is the Procrustes Matching (PM) metric which is defined by quotienting a norm over the group, namely

$$d_{\mathcal{G}_\pm}(\boldsymbol{X}, \boldsymbol{Y}) = \left[ \min_{(\pi, \boldsymbol{R}, t) \in \mathcal{G}_\pm} \sum_{j=1}^n \|x_j - \boldsymbol{R} y_{\pi(j)} + t\|_2^2 \right]^{1/2} = \min_{g \in \mathcal{G}_\pm} \|\boldsymbol{X} - g\boldsymbol{Y}\|_F, \tag{1}$$

where $\|\cdot\|_F$ denote the Frobenius norm. This metric is a generalization of the classical Procrustes metric, where the minimum is only taken over rigid motions, but not over permutations. The

classical Procrustes metric is appropriate for settings where the correspondences between the points $x_i$ and $y_i$ is already known, while the Procrustes Matching metric is appropriate for the unknown correspondence setting. The term Procrustes Matching is taken from Maron et al. (2016).

There have been many works on utilizing the PM metric, and on how to compute it. For example, the celebrated ICP algorithm Besl & McKay (1992) can be seen as optimizing over $\mathcal{G}_\pm$ to compute the PM distance (although the mapping $\pi$ is typically not required to be a permutation). Other papers discussing this include Rangarajan et al. (1997); Maron et al. (2016); Dym & Kovalsky (2019); Dym & Lipman (2017).

If one is interested only in invariance to $\mathcal{G}_+$, a similar metric can be defined by quotienting over $\mathcal{G}_+$, namely
$$d_{\mathcal{G}_+}(\boldsymbol{X}, \boldsymbol{Y}) = \min_{g \in \mathcal{G}_+} \|\boldsymbol{X} - g\boldsymbol{Y}\|_F,$$

**Centralization** A useful fact for later sections, is the simplification of the metrics $d_{\mathcal{G}_\pm}$ and $d_{\mathcal{G}_+}$ for centralized point clouds. For a given $\boldsymbol{X} \in \mathbb{R}^{d \times n}$, centralizing $\boldsymbol{X}$ means translating all its coordinates by the mean $\frac{1}{n}\sum_{j=1}^n x_j$, so that the centralized point cloud now has a mean of $0_d$. It is well known that, for $\boldsymbol{X}, \boldsymbol{Y}$ which are centralized, if $g = (\pi, \boldsymbol{R}, t)$ in $\mathcal{G}_\pm$ (or $\mathcal{G}_+$) is the element for which $\|\boldsymbol{X} - g\boldsymbol{Y}\|_F$ is minimal, then the translation component is zero $t = 0_d$.

**The Hard-Gromov-Wasserstein Metric** An alternative $\mathcal{G}_\pm$ invariant metric can be obtained by comparing pairwise distances. It is known (see e.g., Satorras et al. (2021)), that $\boldsymbol{X}$ and $\boldsymbol{Y}$ are $\mathcal{G}_\pm$ isomorphic as point sets, if and only if the pairwise distances are all the same, namely $\|x_i - x_j\|_2 = \|y_1 - y_j\|_2$ for all $1 \le i < j \le n$. Based on this, we can define a $\mathcal{G}_+$ invariant metric by comparing pairwise distances, and quotienting over permutations, namely

$$d_{\mathcal{D}}(\boldsymbol{X}, \boldsymbol{Y}) = \min_{\pi \in S_n} \sum_{i,j} |\,\|x_i - x_j\|_2 - \|y_{\pi(i)} - y_{\pi(j)}\|_2\,| \tag{2}$$

This metric closely resembles the Gromov-Wasserstein (GW) metric Mémoli (2011), the difference being that the GW metric is defined via minimization over all doubly stochastic matrices, allowing 'soft correspondences', while the metric in equation 2 is defined by minimization over permutations only. For this reason we refer to this metric as the Hard-GW metric. Works focusing on the (Hard or 'soft') GW metric include Chen & Koltun (2015); Solomon et al. (2016); Dym et al. (2017).

We note that the GW metric is much more general: it defines distances between measured metric spaces, up to isometric equivalence. In our setting, we are only considering metric spaces obtained by sampling $n$ points in the Euclidean setting.

## 2.1 Bi-Hölder equivalence of metrics

Now that we have introduced two $\mathcal{G}_\pm$-invariant metrics, we discuss the relationship between them via the notion of Bi-Hölder equivalence. The following theorem shows equivalence with Hölder exponents of $(2, 1)$. After this theorem we will show that the 2 Hölder exponent cannot be improved, and in particular the two metrics are not bi-Lipschitz equivalent.

**Theorem 2.1.** *For all* $\boldsymbol{X}, \boldsymbol{Y} \in \mathbb{R}^{d \times n}$ *we have*

$$d_{\mathcal{D}}(\boldsymbol{X}, \boldsymbol{Y}) \le 2n^{3/2} \cdot d_{\mathcal{G}_\pm}(\boldsymbol{X}, \boldsymbol{Y}).$$

*Additionally, for all $\boldsymbol{X}, \boldsymbol{Y} \in \mathbb{R}^{d \times n}$ satisfying $\|\boldsymbol{X}\|_{1,2} \leq 1, \|\boldsymbol{Y}\|_{1,2} \leq 1$, we have*

$$\frac{1}{4n+2} d^2_{\mathcal{G}_{\pm}}(\boldsymbol{X}, \boldsymbol{Y}) \leq d_{\mathcal{D}}(\boldsymbol{X}, \boldsymbol{Y})$$

We recall that the norm $\|\boldsymbol{X}\|_{1,2}$ in the statement of the theorem, denotes an operator norm with respect to the 1 and 2 norms on the domain and image of $\boldsymbol{X}$, respectively. Equivalently, it is equal to (see Dym (2013))

$$\|\boldsymbol{X}\|_{1,2} = \max_{1 \leq i \leq n} \|x_i\|_2, \tag{3}$$

We note that the choices of norm throughout is just for convenience, we can employ equivalence of norms on finite dimensional spaces to obtain similar results for other norms. However, some sort of boundedness is required to obtain a Hölder inequality. This can be seen from the fact that both metrics are homogenous.

*Proof of Theorem 2.4.* We begin with the first direction of finding **upper Lipschitz bounds**. Let $\boldsymbol{X}, \boldsymbol{Y} \in \mathbb{R}^{d \times n}$. Then for every group element $(\pi, \boldsymbol{R}, t) \in \mathcal{G}_{\pm}$, we have

$$\begin{aligned}
d_{\mathcal{D}}(X, Y) &\leq \sum_{i,j} |\|x_i - x_j\|_2 - \|y_{\pi(i)} - y_{\pi(j)}\|_2| \\
&= \sum_{i,j} |\|x_i - x_j\|_2 - \|\boldsymbol{R}(y_{\pi(i)} - y_{\pi(j)})\|_2| \\
&\leq \sum_{i,j} \|x_i - x_j - \boldsymbol{R}(y_{\pi(i)} - y_{\pi(j)}) - t + t\|_2 \\
&\leq \sum_{i,j} \|x_i - \boldsymbol{R}(y_{\pi(i)}) - t\|_2 + \|x_j - \boldsymbol{R}(y_{\pi(j)}) - t\|_2 \\
&= 2n \cdot \sum_{i} \|x_i - \boldsymbol{R}(y_{\pi(i)}) - t\|_2 \\
&\leq 2n \cdot \sqrt{n} \left[ \sum_{i} \|x_i - \boldsymbol{R}(y_{\pi(i)}) - t\|_2^2 \right]^{1/2}
\end{aligned}$$

Since this is true for all $(\pi, \boldsymbol{R}, t) \in \mathcal{G}_{\pm}$, we can take the minimum over all elements in $\mathcal{G}_{\pm}$ to obtain our first direction, namely $d_{\mathcal{D}}(\boldsymbol{X}, \boldsymbol{Y}) \leq 2n^{3/2} \cdot d_{\mathcal{G}_{\pm}}(\boldsymbol{X}, \boldsymbol{Y})$.

We now consider the second direction of obtaining a **Hölder bound of 2**. Choose any $\boldsymbol{X}, \boldsymbol{Y} \in \mathbb{R}^{d \times n}$, such that $\|\boldsymbol{X}\|_{1,2}, \|\boldsymbol{Y}\|_{1,2} \leq 1$.

Note that since the metrics we are considering are invariant to permutations, we can assume without loss of generality that the permutation minimizing the expression in the definition of $d_{\mathcal{D}}$ is the identity. This implies that

$$d_{\mathcal{D}}(\boldsymbol{X}, \boldsymbol{Y}) = \sum_{i,j} |\|x_i - x_j\|_2 - \|y_i - y_j\|_2| \tag{4}$$

Moreover, since the distances we are interested in are translation invariant, we can translate $\boldsymbol{X}$ by $x_1$ and $\boldsymbol{Y}$ by $y_1$, and so assume without loss of generality that $x_1 = 0 = y_1$. This assumption fixes

the translation degree of freedom, and gives us $\|x_j\|_2 = \|x_j - x_1\|_2$ and $\|y_j\|_2 = \|y_j - y_1\|_2$. Note that the after applying this translation, the 2-norm of each point in $\boldsymbol{X}$ and $\boldsymbol{Y}$ is bounded by 2, and not 1. The distance $\|x_i - x_j\|_2$ is bounded by two as well (the translation does not effect this distance).

The strategy of the proof is to bound the distance between the Gram matrix $\mathcal{G}_\pm(\boldsymbol{X}) = \boldsymbol{X}^T\boldsymbol{X}$ of $\boldsymbol{X}$ and the Gram matrix of $\boldsymbol{Y}$, and then use known results linking the Gram matrix and the Procrustes metric. The entry $(i,j)$ of the Gram matrix of $\boldsymbol{X}$ is the inner product $\langle x_i, x_j \rangle$, and this can be computed via norms and pairwise distances, as

$$\langle x_i, x_j \rangle = \frac{1}{2}\left(\|x_i\|_2^2 + \|x_j\|_2^2 - \|x_i - x_j\|_2^2\right)..$$

Thus, the difference between the entries $(i,j)$ of the Gram matrices is bounded by

$$2|\langle x_i, x_j \rangle - \langle y_i, y_j \rangle| \leq |\,\|x_i\|_2^2 - \|y_i\|_2^2\,| + |\,\|x_i - x_j\|_2^2 - \|y_i - y_j\|_2^2\,| + |\,\|x_j\|_2^2 - \|y_j\|_2^2\,|$$
$$\leq 4|\,\|x_i\|_2 - \|y_i\|_2\,| + 4|\,\|x_i - x_j\|_2 - \|y_i - y_j\|_2\,| + 4|\,\|x_j\|_2 - \|y_j\|_2\,|$$

where for the last inequality, we use the identity

$$\|a\|_2^2 - \|b\|_2^2 = (\|a\|_2 - \|b\|_2)(\|a\|_2 + \|b\|_2),$$

and the fact that by assumption the norms and pairwise differences of all points in $\boldsymbol{X}, \boldsymbol{Y}$ are bounded by 2, as discussed above. Summing over the previous inequality, we obtain a bound on the elementwise-1 norm

$$2\|\boldsymbol{G}(\boldsymbol{X}) - \boldsymbol{G}(\boldsymbol{Y})\|_1 = 2\sum_{i,j}|\langle x_i, x_j \rangle - \langle y_i, y_j \rangle|$$
$$\leq \sum_{i,j}4|\,\|x_i\|_2 - \|y_i\|_2\,| + 4|\,\|x_i - x_j\|_2 - \|y_i - y_j\|_2\,| + 4|\,\|x_j\|_2 - \|y_j\|_2\,|$$
$$\leq 8n\sum_i |\,\|x_i\|_2 - \|y_i\|_2\,| + 4\sum_{i,j}|\,\|x_i - x_j\|_2 - \|y_i - y_j\|_2\,|$$
$$\leq (8n + 4)\sum_{i,j}|\,\|x_i - x_j\|_2 - \|y_i - y_j\|_2\,|$$
$$= (8n + 4)d_\mathcal{D}(\boldsymbol{X}, \boldsymbol{Y}).$$

Next, we can now use the fact that the Gram matrix is positive semi-definite (PSD), and two lemmas taken from previous work. The first lemma is taken from Powers & Størmer (1970):

**Lemma 2.2** (Powers & Størmer (1970), Lemma 4.1)**.** For any two PSD matrices $\boldsymbol{G}_1, \boldsymbol{G}_2$ of the same dimensions, it holds that

$$|\sqrt{\boldsymbol{G}_1} - \sqrt{\boldsymbol{G}_2}|_F^2 \leq \|\boldsymbol{G}_1 - \boldsymbol{G}_2\|_T.$$

Here $\|\cdot\|_F$ denotes the Frobenius norm (the 2-norm of the matrix singular values) and $\|\cdot\|_T$ denotes the trace norm, or nuclear norm (the 1-norm of the matrix singular values). It is known that this norm is always smaller or equal to the elementwise-1 norm.

The second lemma is taken from Derksen (2024)

**Lemma 2.3** (Theorem 3 in Derksen (2024)). *For all $\boldsymbol{X}, \boldsymbol{Y} \in \mathbb{R}^{d \times n}$,*

$$\min_{\boldsymbol{R} \in O(d)} \|\boldsymbol{R}\boldsymbol{X} - \boldsymbol{Y}\|_F \leq \|\sqrt{\boldsymbol{G}(\boldsymbol{X})} - \sqrt{\boldsymbol{G}(\boldsymbol{Y})}\|_F \leq \sqrt{2} \min_{\boldsymbol{R} \in O(d)} \|\boldsymbol{R}\boldsymbol{X} - \boldsymbol{Y}\|_F.$$

Piecing all of this together, we obtain for all $\boldsymbol{X}, \boldsymbol{Y}$ such that $\|\boldsymbol{X}\|_{1,2}, \|\boldsymbol{Y}\|_{1,2} \leq 1$, our desired inequality

$$d_{\mathcal{G}_\pm}^2(\boldsymbol{X}, \boldsymbol{Y}) \leq \min_{\boldsymbol{R} \in O(d)} \|\boldsymbol{R}\boldsymbol{X} - \boldsymbol{Y}\|_F^2 \leq \|\sqrt{\boldsymbol{G}(\boldsymbol{X})} - \sqrt{\boldsymbol{G}(\boldsymbol{Y})}\|_F^2$$

$$\leq \|\boldsymbol{G}(\boldsymbol{X}) - \boldsymbol{G}(\boldsymbol{X})\|_T \leq \|\boldsymbol{G}(\boldsymbol{X}) - \boldsymbol{G}(\boldsymbol{X})\|_1 \leq (4n + 2)d_{\mathcal{D}}(\boldsymbol{X}, \boldsymbol{Y})$$

$\square$

We next show that the Hölder exponent of 2 in the previous theorem cannot be improved:

**Theorem 2.4.** *For natural numbers $n \geq 3, d \geq 2$, let $c, \alpha > 0$ such that, for all $\boldsymbol{X}, \boldsymbol{Y} \in \mathbb{R}^{d \times n}$ satisfying $\|\boldsymbol{X}\|_{1,2} \leq 1, \|\boldsymbol{Y}\|_{1,2} \leq 1$, we have*

$$c d_{\mathcal{G}_\pm}^\alpha(\boldsymbol{X}, \boldsymbol{Y}) \leq d_{\mathcal{D}}(\boldsymbol{X}, \boldsymbol{Y}). \tag{5}$$

*Then $\alpha \geq 2$.*

*Proof.* Let $c, \alpha$ be positive numbers such that equation 5 is satisfied. Choose $a_1 = 0_{d-1} \in \mathbb{R}^{d-1}$ and $a_2, \ldots, a_n \in \mathbb{R}^{d-1}$ such that

1. $a_1, \ldots, a_n$ are pairwise distinct

2. $1 > \|a_j\|, \quad \forall j = 1, \ldots, n$ and

3. $\sum_{j=1}^n a_j = 0$.

Such a choice is possible as long as $n \geq 3$. For $\epsilon \geq 0$ denote

$$\boldsymbol{X}^\epsilon = \begin{bmatrix} a_1 = 0_{d-1} & a_2 & a_3 & \ldots & a_n \\ \epsilon & -\epsilon & 0 & \ldots & 0 \end{bmatrix}$$

and denote $\boldsymbol{X} = \boldsymbol{X}^0$. For all $\epsilon$ small enough, the norms of the columns of $\boldsymbol{X}^\epsilon$ are all less than one, and so are in the domain $\|\boldsymbol{X}^\epsilon\|_{1,2} \leq 1$ which we are considering.

Next, we claim that for all $\epsilon > 0$ small enough we have $d_{\mathcal{G}_\pm}(\boldsymbol{X}, \boldsymbol{X}^\epsilon) \geq \epsilon$. To see this, note that $\boldsymbol{X}^\epsilon$ and $\boldsymbol{X}$ are centralized, and therefore the optimal translation is zero. If $(\pi, \boldsymbol{R})$ are the optimal permutation-rotation pair aligning $\boldsymbol{X}$ with $\boldsymbol{X}^\epsilon$, and the column $x_1^\epsilon$ of $\boldsymbol{X}^\epsilon$ is assigned to the column $x_{\pi(1)}$ of $\boldsymbol{X}$, then

$$d_{\mathcal{G}_\pm}(\boldsymbol{X}, \boldsymbol{X}^\epsilon) \geq \|x_1^\epsilon - \boldsymbol{R}x_{\pi(1)}\|_2 \geq \big| \|x_1^\epsilon\|_2 - \|x_{\pi(1)}\|_2 \big| = |\epsilon - \|a_{\pi(1)}\|_2|.$$

if $\pi(1) \neq 1$, then for all small enough $\epsilon$, this expression will be larger than $\epsilon$ since $a_{\pi(1)} \neq a_1 = 0_{d-1}$. If $\pi(1) = 1$, then this expression will be exactly $\epsilon$. In any case, we obtain that the distance between $\boldsymbol{X}$ and $\boldsymbol{X}^\epsilon$ cannot be less than $\epsilon$.

Next, we consider $d_{\mathcal{D}}(\boldsymbol{X}^\epsilon, \boldsymbol{X})$. We note that $\|x_i - x_j\|_2 = \|x_i^\epsilon - x_j^\epsilon\|_2$ when both $i, j$ are larger than 2. If $i \in \{1, 2\}$ and $j > 2$, then for all $\epsilon$ with $\frac{|\epsilon|}{\|a_i - a_j\|_2} \leq 1/2$, we have an inequality of the form

$$| \|x_i - x_j\|_2 - \|x_i^\epsilon - x_j^\epsilon\|_2 | = | \|a_i - a_j\|_2 - \sqrt{\|a_i - a_j\|_2^2 + \epsilon^2} |$$

$$= \|a_i - a_j\|_2 \left( 1 - \sqrt{1 + \frac{\epsilon^2}{\|a_i - a_j\|_2^2}} \right)$$

$$\leq C \frac{\epsilon^2}{\|a_i - a_j\|_2}$$

for an appropriate $C$ which can be derived from the Taylor expansion of $\sqrt{1 + x}$ around $x = 0$. A similar bound can be obtained when $i = 1, j = 2$, and so overall we can show that such a bound holds for all $i, j$. It follows that for all small enough $\epsilon > 0$, there exists some $\tilde{C}$ which does not depend on $\epsilon$, such that

$$c\epsilon^\alpha \leq cd_{\mathcal{G}_\pm}^\alpha(\boldsymbol{X}, \boldsymbol{X}^\epsilon) \leq d_{\mathcal{D}}(\boldsymbol{X}, \boldsymbol{X}^\epsilon) \leq \tilde{C}\epsilon^2.$$

Since this inequality holds for all small enough positive $\epsilon$, this inequality can only hold if $\alpha \geq 2$. $\qquad\square$

## 3   1-WL Geometric Models and bi-Lipschitzness

Now that we have discussed $\mathcal{G}_\pm$ invariant metrics on $\mathbb{R}^{d \times n}$, we turn to discuss $\mathcal{G}_\pm$ invariant models on $\mathbb{R}^{d \times n}$, and the challenge of designing them so that they are bi-Lipschitz equivalent to the metric $d_{\mathcal{G}_\pm}$.

Our focus will be on models which apply graph neural networks to pairwise features which are invariant to rigid motions. Namley, each point set in $\mathbb{R}^{d \times n}$ is represented by a weighted full graph with $n$ nodes, where all pairs of nodes are assigned a weight $\|x_i - x_j\|_2$. Since these weights are invariant to rigid motions, it can be shown Lim et al. (2022); Li et al. (2023) that one can obtain $\mathcal{G}_\pm$ invariant neural networks by applying permutation invariant neural networks to this weighted graph.

Among the most popular graph neural networks are the family of message passing neural networks, which iteratively update node features by aggregating information over node neighbors. This procedure can be adapted to the geometric point set setting in the following way: we first define an initial 'blank' feature for each node, namely, $c_i^0 = 0$ for all $i \in [n]$. Next, for all $t = 0, \ldots, T-1$, where $T$ is the number of iterations prescribed by the 'user', we define

$$c_i^{t+1} = \phi_t(c_i^t, \psi_t(\{\!\{(c_i^t, c_j^t, \|x_i - x_j\|_2) \mid j \in [n]\}\!\})).$$

Here, $\psi_t$ is a function applied to a *multiset*, which means that is needs to be invariant to the order in which the points are arranged. This is required to ensure permutation invariance of the procedure. After $T$ iterations are concludes, a final $\mathcal{G}_\pm$ invariant global feature $c_{\text{global}}$ is obtained by applying an additional multiset function, call ReadOut, to the final features of all nodes, namely

$$c_{\text{global}} = \text{ReadOut}(\{\!\{c_1^T, \ldots, c_n^T\}\!\}). \tag{6}$$

We will denote the function obtained by such a procedure with $T$ iterations, by

$$\Phi_T(\boldsymbol{X}) = c_{\text{global}}(\boldsymbol{X})$$

We name models which are built in this way *1-WL geometric models* (the source to this name is the connection between the expressive power of MPNNs and the 1-WL graph isomorphism test).

There are many popular geometric models which fall under the category of 1-WL geometric models. Examples include SchNet Schütt et al. (2018), SphereNet Liu et al. (2021),DimeNet Gasteiger et al. (2020b) and EGNN Satorras et al. (2021). For example, in (the invariant version of) EGNN the feature update procedure is given by

$$c_i^{t+1} = \phi_t \left( c_i^t, \sum_j \eta_t(c_i^t, c_j^t, \|x_i - x_j\|_2^2) \right),$$

where $\phi_t$ and $\eta_t$ are fully connected neural networks, whose internal paremters are optimized in accordance with the specified learning problem. In other words, EGNN enforces the requirements that $\psi_t$ is a multiset function by applying the same function $\eta_t$ to all multiset elements, and then summing over all results.

A 1-WL geometric model, as defined above, is always $\mathcal{G}_\pm$ invariant. An important question is whether it is complete, in the sense of Definition 1.4. Namely, can the model be designed so that $\Phi_T(\boldsymbol{X}) = \Phi_T(\boldsymbol{Y})$ if and only if $\boldsymbol{X}$ and $\boldsymbol{Y}$ belong to the same $\mathcal{G}_\pm$ orbit?

This question has been studied in recent work. In Pozdnyakov & Ceriotti (2022); Li et al. (2023) it is shown that for $d = 3$, these models are not complete (although they are 'generically complete', see Hordan et al. (2023); Li et al.. However, they are complete when $d = 2$, as shown in Delle Rose et al. (2023). Since completeness is a prerequisite to bi-Lipschitzness, the question of bi-Lipschitzness for these models is only relevant for $d = 2$. We will now address this issue.

Our first result will be that 1-WL geometric models are Lipschitz with respect to the $d_\mathcal{D}$ distance. As a corollary, we will see that they cannot be lower-Lipschitz with respect to $d_{\mathcal{G}_\pm}$, our prime metric of interest. This result employs the assumption that the functions $\phi_t, \psi_t$ and the ReadOut functions are all Lipschitz, this assumption is essentially always fulfilled in practice (see Davidson & Dym (2024) for discussion). We note that when we say that the multiset functions, ReadOut and $\psi_t$, are Lipschitz, this can be interperted as saying that we think of them as functions on matrices whose columns are the multiset elements, and Lipschitzness is understood in the standard Euclidean sense. Equivalently, this can also be thought of as Lipschitzness in the sense of the Wasserstein distance on multisets (introduced later on).

**Theorem 3.1.** *For given natural $n, d$ and $T$, let $\Phi_T$ be a 1-WL geometric models, and assume that the functions $\phi_t, \psi_t$ and ReadOut are upper Lipschitz. Then $\Phi_T$ is Lipschitz with respect to the metric $d_\mathcal{D}$.*

*Proof.* Firstly, since $\phi_T$ is permutation invariant, we can assume without loss of generality that we are only interested in pairs $\boldsymbol{X}, \boldsymbol{Y}$ such that the permutation for which the minimum is obtain in the definition of $d_\mathcal{D}$ is the identity, so that

$$d_\mathcal{D}(\boldsymbol{X}, \boldsymbol{Y}) = \sum_{i,j} |\, \|x_i - x_j\|_2 - \|y_i - y_j\|_2 \,|$$

The claim is more transparent if the iterative process is rewritten as a two step procedure, namely

$$q_i^{t+1} = \psi_t(\{\!\!\{(c_i^t, c_j^t, \|x_i - x_j\|_2) \mid j \in [n]\}\!\!\}))$$
$$c_i^{t+1} = \phi_t(c_i^t, q_i^{t+1})$$

Since $\psi_1$ is Lipschitz and $c_i^0, c_j^0 = 0$, the feature $q_i^{t+1}$ is bi-Lipschitz in its $n^2$ dimensional input. Similarly, one can show iteratively that the features $c_i^t, q_i^t$ are a Lipschitz function of the input for all $t = 1, \ldots, T$. Similarly, $c_{\text{global}}$ is a Lipschitz function of the features $c_i^T$, as defined in equation 6. Therefore, we deduce that for an appropriate $L > 0$,

$$\begin{aligned} \|\Phi_T(\boldsymbol{X}) - \Phi_T(\boldsymbol{Y})\|_2 &= \|c_{\text{global}}(\boldsymbol{X}) - c_{\text{global}}(\boldsymbol{Y})\|_2 \\ &\leq L \sum_{i,j} |\, \|x_i - x_j\|_2 - \|y_i - y_j\|_2 \,| \\ &= L d_{\mathcal{D}}(\boldsymbol{X}, \boldsymbol{Y}). \end{aligned}$$

$\square$

As a corollary of this result and Theorem 2.4, we obtain that a 1-WL geometric model cannot be lower-Lipschitz with respect to $d_{\mathcal{G}_\pm}$:

**Corollary 3.2.** *If $n \geq 3, d \geq 2$, and $T$ is any natural number, and $\Phi_T$ is a 1-WL geometric model defined via Lipschitz functions as in Theorem 3.1, then $\Phi_T$ cannot be lower Lipschitz with respect to $d_{\mathcal{G}_\pm}$.*

*Proof.* By the previous thoerem we know that $\Phi_T$ is upper Lipschitz with respect to $d_{\mathcal{D}}$. Assume by contradiction that $\Phi_T$ were also lower Lipschitz with respect to $d_{\mathcal{G}_\pm}$. Then for appropriate constants $c, C > 0$ we would have

$$c \cdot d_{\mathcal{G}_\pm}(\boldsymbol{X}, \boldsymbol{Y}) \leq \|\Phi_T(\boldsymbol{X}) - \Phi_T(\boldsymbol{Y})\|_2 \leq C \cdot d_{\mathcal{D}}(\boldsymbol{X}, \boldsymbol{Y}) \quad \forall \boldsymbol{X}, \boldsymbol{Y} \in \mathbb{R}^{d \times n}.$$

which is a contradiction to Theorem 2.4, that states than an inequality of the form $c/C \cdot d_{\mathcal{G}_\pm}^\alpha(\boldsymbol{X}, \boldsymbol{Y}) \leq d_{\mathcal{D}}(\boldsymbol{X}, \boldsymbol{Y})$ can only hold for $\alpha \geq 2$. $\square$

We note that while this claim holds for $d \geq 2$, the main novelty is the case $d = 2$ where 1-WL geometric models can be complete. As mentioned previously, when $d > 2$ 1-WL geometric models are not complete, and thus the corollary also follows from the fact that there are pairs $\boldsymbol{X}, \boldsymbol{Y}$ which are not in the same orbit, and hence have positive distance, but for which $\|\Phi_T(\boldsymbol{X}) - \Phi_T(\boldsymbol{Y})\|_2 = 0$.

## 4  Bi-Lipschitz point set models

In the previous section, we showed that 1-WL geometric models cannot be bi-Lipschitz, even in the case where $d = 2$ and they are complete. In this section, we explain how to obtain models which are bi-Lipschitz in the $d = 2$ setting.

The model we propose will be similar in structure to geometric 1-WL models. It will be bi-Lipschitz even with a single iteration $T = 1$, and for simplicity of exposition, we will define the model in

this case only, as extending it to general $T$ can be done in a straightforward fashion. The model is defined by

$$\text{centralize } \boldsymbol{X} \tag{7}$$

$$h_i^{(1)}(\boldsymbol{X}) = \psi\left( \|x_i\|_2, \left\{\!\!\left\{ \left( \frac{x_i \cdot x_j}{\|\boldsymbol{X}\|_F}, \frac{x_i^\perp \cdot x_j}{\|\boldsymbol{X}\|_F}, \|x_j\|_2 \right) \mid j \in [n] \setminus \{i\} \right\}\!\!\right\} \right) \tag{8}$$

$$H(\boldsymbol{X}) = \phi(\{\!\!\{ h_i^{(1)} \mid i \in [n] \}\!\!\}) \tag{9}$$

where for a vector $x_i = [a, b] \in \mathbb{R}^2$ we define $x_i^\perp$ to be the perpendicular vector $x_i^\perp = [-b, a]$. The idea to include this vector comes from the fact that (unless $x_i = 0_2$), the vectors $x_i$ and $x_i^\perp$ form a basis for $\mathbb{R}^2$, and the inner products of any vector $x$ with these two basis elements determines $x$ uniquely. A similar idea is was suggested in Hordan et al. (2023).

$\mathcal{G}_+$ **invariance**  We note that the construction above is $\mathcal{G}_+$ invariant but not $\mathcal{G}_\pm$ invariant, this is because the function

$$\mathbb{R}^{2 \times 2} \ni (x_i, x_j) \mapsto x_i^\perp \cdot x_j$$

is $SO(2)$ invariant but not $O(2)$ invariant (as it depends on the choice of the direction of $x_i^\perp$) . As we will see later on, this construction is $\mathcal{G}_+$ complete, and bi-Lipschitz with respect to $d_{\mathcal{G}_+}$. The fact that these results holds for $\mathcal{G}_+$ rather than $\mathcal{G}_\pm$ is, in our opinion, an advantage. On the one hand, in several problems the orientation may be important, and $O(d)$ invariance is not desired in this case (e.g., the concept of chirality in chemistry applications, see Gaiński et al. (2023)). On the other hand, a $\mathcal{G}_+$ bi-Lipschitz model can easily be transformed into a $\mathcal{G}_\pm$ bi-Lipschitz model by a simple procedure of symmetrization via a single reflection. This procedure is discussed in Appendix A.

**Requirements on $\phi$ and $\psi$**  To achieve bi-Lipschitzness, we will need to choose the functions $\psi$ and $\phi$ correctly. We will require that both these functions are (a) homogenous and (b) bi-Lipschitz.

By homogeniety we mean that, for a multiset $S$ and vector[1]$a$,

$$\phi(tS) = s\phi(S), \quad \psi(ta, tS) = t\psi(a, S), \quad \forall t > 0.$$

Next, we discuss what we mean by bi-Lipschitzness of $\phi$ and $\psi$.

The readout function $\phi$ maps multisets to vectors, and so it is required to be be bi-Lipschitz with respect to a $p$ norm on the vector space, and a $p$-Wasserstein distance on the multiset space. The exact choice of $p$ does not matter to the question of bi-Lipschitzness, due to equivalence of $p$-norms. For convenience, in the analysis later on we will consider $p = \infty$, where the $\infty$-Wasserstein distance is defined via

$$\mathcal{W}_\infty\left(S, S'\right) \triangleq \min_{\tau \in S_n} \max_{i \in [n]} \|x_i - y_{\tau(i)}\|_\infty. \tag{10}$$

Thus, our requirement for $\phi$ is that for given mutlisets $S, S'$, there exist constants $0 < c_\phi \le C_\phi$ suc that

$$c_\phi \mathcal{W}_\infty\left(S, S'\right) \le \|\phi(S) - \phi(S')\|_\infty \le C_\phi \mathcal{W}_\infty\left(S, S'\right)$$

---

[1]in the case $d = 2$ we are currently discussing, $a$ is actually a scalar. However, later on when we discuss $d > 2$ we will have an analogous construction where $a$ is a vector.

There are several possible choices of multiset-to-vector functions which are both bi-Lipschitz and homogenous. One popular choice, introduced by Balan et al. (2022); Dym & Gortler (2024), is the multi-set function $\beta$ whose coordinates are given by

$$\beta_i\left(\{\!\{x_1, \ldots, x_n\}\!\}\right) = b_i \cdot \text{sort}\left(a_i \cdot x_1, \ldots, a_i \cdot x_n\right), \quad i \in \{1, 2, \ldots, m\},$$

where $b_i \in \mathbb{R}^n, a_i \in \mathbb{R}^d$ are parameters of the function. It was shown in Balan et al. (2024) that for $m \geq 2nd + 1$, where $d$ is the dimension of the points in the multisets, and for generic parameters $a_i, b_i$ of $\beta$, it will be bi-Lipschitz with respect to the Wasserstein distance. This bound was recently improved by Dym et al. (2025b) to $m \geq (2n - 1)d$ Additionally, this function is clearly homogenous. Other examples of homogenous bi-Lipschitz functions is the FSW embedding from Amir & Dym and the max filters from Cahill et al. (2025). In our experiment we used the function $\beta$ described above.

Next, we discuss the bi-Lipchitzness of the function $\psi$. This function maps a vector-multiset pair $(v, S)$ to a vector $y$. We would like this function to be bi-Lipschitz, in the sense that there exist constant $0 < c_\psi \leq C_\psi$ such that

$$c_\psi \max\{\|v - v'\|_\infty, \mathcal{W}_\infty\left(S, S'\right)\} \leq \|\psi(v, S) - \psi(v', S')\|_\infty \leq C_\psi \max\{\|v - v'\|_\infty, \mathcal{W}_\infty\left(S, S'\right)\}$$

To obtain a function $\psi$ which is both homogenous and bi-lipschitz, we can choose the function $\beta$ defined previously (or any other homogenous bi-Lipschitz function on multisets) and extend it to a homogenous bi-Lipschitz function on vector-multisets pairs simply by

$$\psi(v, S) \triangleq (v, \beta(S)).$$

**Singularity at zero**   The model we describe is apriori not defined at $\boldsymbol{X} = 0_{d \times n}$, since we divide by the norm of $\boldsymbol{X}$. However, as we require that $\phi$ and $\psi$ are homogenous functions, we obtain as a result that $H(\boldsymbol{X})$ is homogenous, that is $H(t\boldsymbol{X}) = tH(\boldsymbol{X})$ for all $t > 0$. This naturally leads to defining

$$H(0_{d \times n}) = 0_m,$$

where $m$ is the output dimension of $H$. With this definition, $H$ is bi-Lipschitz on the whole domain $\mathbb{R}^{2 \times n}$, as the following theorem shows:

**Theorem 4.1.** *If $\psi$ and $\phi$ are bi-Lipschitz and homogenous, as discussed above, Then $H : \mathbb{R}^{2 \times n} \to \mathbb{R}^m$ is bi-Lipschitz with respect to the metric $d_{\mathcal{G}_\pm}$.*

*Proof.* This theorem is a special case of our theorem for general $d \geq 2$, Theorem 3.1, which will be stated and proven below. □

### 4.1   A bi-Lipschitz (d-1)-WL embedding for $d$-dimensional point clouds

We now generalize the construction from the previous section to derive a $\mathcal{G}_+$ invariant model for the case of point sets in $\mathbb{R}^d, d \geq 3$, which is bi-Lipschitz with respect to the metric $d_{\mathcal{G}_+}$. We will first explicitly discuss the case $d = 3$ which is the most common in applications.

To achieve bi-Lipschitzness, a prerequisite is completeness. As discussed previously, it is known that geometric 1-WL models, and similar models based on propagation of node features, are complete for

point sets with $d = 2$, but not for point sets with $d \geq 3$. In the latter case, a more computationally demanding procedure of storing a feature per each pair of points, is needed for completeness Delle Rose et al. (2023); Hordan et al. (2023); Widdowson & Kurlin (2023). We will call this type of models *2-WL geometric models*. The basic idea is that if $x_1, x_2$ are two vectors in $\mathbb{R}^3$ which are linearly independent, then together with their vector product $x_{1,2}^{\perp} := x_1 \times x_2$ they form a basis for $\mathbb{R}^3$, and so inner products of a vector $x$ with the vectors $x_1, x_2$ and $x_1 \times x_2$ define $x$ uniquely (note however, that the completeness does not rely on the assumption that there exists linearly independent $x_1, x_2$).

To achieve completeness, and bi-Lipschitzness, we suggest the following model, which can be seen as a natural generalization of our previous model to the case $d = 3$, and a homogenous version of the model suggested in Hordan et al. (2023). Given a point set $\boldsymbol{X} = \{x_1, \ldots, x_n\}$, our model will be of the form:

$$\text{centralize } \boldsymbol{X} \tag{11}$$

$$h_{ij}^{(1)}(\boldsymbol{X}) = \psi\left(G_{ij}(\boldsymbol{X}), \left\{\!\!\left\{\left(\frac{x_i \cdot x_k}{\|\boldsymbol{X}\|_F}, \frac{x_j \cdot x_k}{\|\boldsymbol{X}\|_F}, \frac{(x_i \times x_j) \cdot x_k}{\|\boldsymbol{X}\|_F^2}, \|x_k\|_2\right) \mid k \in [n]/\{i,j\}\right\}\!\!\right\}\right) \tag{12}$$

$$H(\boldsymbol{X}) = \phi(\{\!\!\{h_{ij}^{(1)} \mid i \neq j \in [n]\}\!\!\}), \tag{13}$$

where $G_{ij}(\boldsymbol{X})$ is the $2 \times 2$ symmetric matrix

$$G_{ij}(\boldsymbol{X}) = \begin{pmatrix} \|x_i\|_2 & \frac{x_i \cdot x_j}{\|\boldsymbol{X}\|_F} \\ \frac{x_i \cdot x_j}{\|\boldsymbol{X}\|_F} & \|x_j\|_2, \end{pmatrix},$$

To define our construction for a general dimension $d$, we need to introduce some notation. We will assume that $d \leq n$. Let $T(n, d-1)$ be the set of all $(d-1)$-tuples of unique integers in $\{1, \ldots, n\}$ sorted in ascending order. We will use the following abbreviated notation for multi-indices in $T(n, d-1)$:

$$\mathbf{i} = (i_1, \ldots, i_{d-1}) \in T(n, d-1).$$

We define $G_{\mathbf{i}}$ to be the $(d-1) \times (d-1)$ normalized Gram matrix defined as

$$G_{\mathbf{i}}(\boldsymbol{X})[m, \ell] = \begin{cases} \frac{x_m \cdot x_\ell}{\|X\|_F} & m \neq \ell \\ \|x_m\|_2 & m = \ell \end{cases}, \tag{14}$$

Finally, for each such collection of $d - 1$ indices $\mathbf{i}$, corresponding to $d - 1$ vectors $x_{i_1}, \ldots, x_{i_{d-1}}$, we define a unique vector $x_{\mathbf{i}}^{\perp}$ which is orthogonal to those vectors: the Hodge-dual of the wedge-product $x_{i_1} \bigwedge x_{i_2} \cdots \bigwedge x_{i_{d-1}}$ Darling (1994). The latter, is a coordinate-free definition of a generalized cross product in $\mathbb{R}^d$, and can be computed as the unique vector $z \in \mathbb{R}^d$ such that

$$\langle z, y \rangle = \det(x_{i_1} \cdots x_{i_{d-1}} \ y), \quad y \in \mathbb{R}^d, \tag{15}$$

where $z$ is the unique vector dual (guaranteed by Riesz's representation theorem) of the functional defined by the r.h.s of equation 15, and we let $y$ vary. From this definition we easily see that (i) the vector $z$ is orthogonal to all of the vectors $x_{i_j}, j = 1, \ldots, d-1$ and (ii) the coordinates of $z$ are $(d-1)$-homogenous polynomials in the vectors $x_{i_j}, j = 1, \ldots, d-1$.

the $\mathcal{G}_+$ invariant model $H$ we suggest, is defined for a given point set $\boldsymbol{X} = \{x_1, \ldots, x_n\} \subset \mathbb{R}^d$ where $n \geq d$, by

$$\text{centralize } \boldsymbol{X} \tag{16}$$

$$h_{\mathbf{i}}^{(1)}(\boldsymbol{X}) = \psi(G_{\mathbf{i}}, \{\!\{ \left( \frac{x_{i_1} \cdot x_k}{\|\boldsymbol{X}\|_F}, \ldots, \frac{x_{i_{d-1}} \cdot x_k}{\|\boldsymbol{X}\|_F}, \frac{x_{\mathbf{i}}^\perp \cdot x_k}{\|\boldsymbol{X}\|_F^{d-1}}, \|x_k\|_2 \right) \mid k \in [n]/\{i_1, \ldots, i_{d-1}\} \}\!\}) \tag{17}$$

$$H(\boldsymbol{X}) = \phi(\{\!\{ h_{\mathbf{i}}^{(1)} \mid \mathbf{i} \in T(n, d-1) \}\!\}), \tag{18}$$

In the cases $d = 3$ and $d = 2$ this reduces to the models defined in the previous subsections.

To achieve bi-Lipschitzness with this model, we will assume, as in the case $d = 2$, that $\Psi$ and $\Phi$ are homogenous and Bi-Lipschitz as described above. Also as before, we will make the natural extension $H(0_{d \times n}) = 0_m$. For the proof we will need to make an assumption on our domain. Namely, instead of considering all of $\mathbb{R}^{d \times n}$ as we did in the $d = 2$ case, we fix a positive $c > 0$ and consider point sets which, after centralization, reside in

$$\Omega_c = \{ \boldsymbol{X} \in \mathbb{R}^{d \times n} \mid \boldsymbol{X} \text{ is centralized and } \max_{\mathbf{i} \in T(n, d-1)} \|x_{\mathbf{i}}^\perp\|_2 \geq c \|\boldsymbol{X}\|_F^{d-1} \}.$$

We will show that the model $H$ defined above will be bi-Lipschitz on any such set. We note that in the special case where $d = 2$, the set $\Omega_c$ equals $\mathbb{R}^{2 \times n}$ for any $c \leq 1$, since when $d = 1$ we have that $\|x_i^\perp\|_2 = \|x_i\|_2$. In particular, we obtain our theorem for the case $d = 2$ as a special case.

**Theorem 4.2.** *For fixed $c > 0$ and natural numbers $n \geq d$, if $\phi, \psi$ are bi-Lipschitz and homogenous, then the restriction of $H$ to $\Omega_c$ is bi-Lipschitz with respect to the metric $d_{\mathcal{G}_+}$.*

*Proof.* Firstly, we note that due to translation invariance of both $H$ and the metric $d_{\mathcal{G}_+}$, it is sufficient to prove the claim for pairs $\boldsymbol{X}, \boldsymbol{Y}$ which are already centralized. This simplifies notation.

**Step 1:** We will begin by proving bi-Lipschitzness in the case where $(\boldsymbol{X}, \boldsymbol{Y})$ are both in the compact set

$$K_c = \{ \boldsymbol{X} \in \Omega_c \mid 1/2 \leq \|\boldsymbol{X}\|_F \leq 1 \}.$$

**Step 1(a): Upper Lipschitzness on $K_c$.** The set $K_c$ is contained in the open subtset $U_c \subseteq \mathbb{R}^{d \times n}$ defined as

$$U_c = \{ \boldsymbol{X} \in \mathbb{R}^{d \times n} \mid \|\boldsymbol{X}\|_F > 1/4 \text{ and } \max_{\mathbf{i} \in T(n, d-1)} \|x_{\mathbf{i}}^\perp\|_2 > c/2 \cdot \|\boldsymbol{X}\|_F^{d-1} \}.$$

We also note that the set $K_c$ is closed under rotations and permutations.

Now, note that equation 17 used to define $h_{\mathbf{i}}^{(1)}(\boldsymbol{X})$ can be rewritten as

$$h_{\mathbf{i}}^{(1)}(\boldsymbol{X}) = \psi\left(F_{\mathbf{i}}(\boldsymbol{X})\right), \text{ where}$$

$$F_{\mathbf{i}}(\boldsymbol{X}) = \left( G_{\mathbf{i}}(\boldsymbol{X}), \frac{x_{i_1} \cdot x_k}{\|\boldsymbol{X}\|_F}, \ldots, \frac{x_{i_{d-1}} \cdot x_k}{\|\boldsymbol{X}\|_F}, \frac{x_{\mathbf{i}}^\perp \cdot x_k}{\|\boldsymbol{X}\|_F^{d-1}}, \|x_k\|_2 \mid k \in [n]/\{i_1, \ldots, i_{d-1}\} \right). \tag{19}$$

We note that $F_{\mathbf{i}}$ is a smooth function with no singularities in $U_c$, therefore, it is Lipschitz in the compact set $K_c \subseteq U_c$. Similarly, by assumption $\psi$ and $\phi$ are Lipschitz. We deduce that the function $H(\boldsymbol{X})$ is Lipschitz, so there exists some $L > 0$ such that

$$\|H(\boldsymbol{X}) - H(\boldsymbol{Y})\|_\infty \leq L \|\boldsymbol{X} - \boldsymbol{Y}\|_F, \quad \forall \boldsymbol{X}, \boldsymbol{Y} \in K_c.$$

For given $\boldsymbol{X}, \boldsymbol{Y} \in K_c$, let $g \in \mathcal{G}_+$ such that $d_{\mathcal{G}_+}(\boldsymbol{X}, \boldsymbol{Y})$ is equal to $\|\boldsymbol{X} - g\boldsymbol{Y}\|_F$. We know that the translation component of $g$ is zero, and that $K_c$ is closed to rotations and permutation. Then, due to the $\mathcal{G}_+$ invariance of $H$, we have

$$\|H(\boldsymbol{X}) - H(\boldsymbol{Y})\|_\infty = \|H(\boldsymbol{X}) - H(g\boldsymbol{Y})\|_\infty \leq L\|\boldsymbol{X} - g\boldsymbol{Y}\|_F = Ld_{\mathcal{G}_+}(\boldsymbol{X}, \boldsymbol{Y}).$$

Thus, we have shown upper Lipschitzness on $K_c$.

**Step 1(b): Lower Lipschitzness on** $K_c$ To show lower Lipschitzness on $K_c$, choose some arbitrary $\boldsymbol{X}, \boldsymbol{Y} \in K_c$, and denote $\epsilon = \mathcal{W}_\infty(H(\boldsymbol{X}), H(\boldsymbol{Y}))$. Our goal is to show that $d_{\mathcal{G}_+}(\boldsymbol{X}, \boldsymbol{Y}) \geq C\epsilon$ for some constant $C$ uniformly over all $\boldsymbol{X}, \boldsymbol{Y} \in K_c$. Since $\boldsymbol{X}$ is in $K_c$, there exists some $\mathbf{i}$ for which

$$\|x_\mathbf{i}^\perp\|_2 \geq c\|\boldsymbol{X}\|_F^{d-1} \geq \frac{c}{2^{d-1}}. \tag{20}$$

Without loss of generality, by permuting the indices of $\boldsymbol{X}$ if necessary, we can assume that $\mathbf{i} = (1, 2, \ldots, d-1)$. We also note that , by equation 18 and equation 10, there exists some $\mathbf{i}'$ such that

$$c_\phi\|h_\mathbf{i}^{(1)}(\boldsymbol{X}) - h_{\mathbf{i}'}^{(1)}(\boldsymbol{Y})\|_\infty \leq \epsilon. \tag{21}$$

We can now permute the indices of $\boldsymbol{Y}$ if necessary so that this equation holds for $\mathbf{i}' = \mathbf{i}$. We next claim that, without loss of generality, we can assume that

$$\|y_\mathbf{i}^\perp\|_2 > \frac{c}{2^d}. \tag{22}$$

Indeed, if this were not the case, and $\|y_\mathbf{i}^\perp\|_2 \leq \frac{c}{2^d}$, then we on the compact set

$$Q = \{(x_\mathbf{i}, y_\mathbf{i})| \quad \|x_\mathbf{i}\|_F, \|y_\mathbf{i}\|_F \leq 1 \text{ and } \|x_\mathbf{i}^\perp\|_2 > \frac{c}{2^{d-1}} \geq \frac{c}{2^d} \geq \|y_\mathbf{i}^\perp\|_2\}$$

we would have that the minimum

$$\eta := \min_{(x_\mathbf{i}, y_\mathbf{i}) \in Q} \|G_\mathbf{i}(\boldsymbol{X}) - G_\mathbf{i}(\boldsymbol{Y})\|_\infty$$

is obtained for some $\eta > 0$ (since if the gram matrices were identical, the vectors $x_\mathbf{i}^\perp, y_\mathbf{i}^\perp$ we would have the same norm. Accordingly, we would have that

$$c_\phi \cdot \eta \leq c_\phi\|h_\mathbf{i}^{(1)}(\boldsymbol{X}) - h_{\mathbf{i}'}^{(1)}(\boldsymbol{Y})\|_\infty \leq \epsilon,$$

and as a result we would have a bi-Lipschitz inequality

$$d_{\mathcal{G}_+}(\boldsymbol{X}, \boldsymbol{Y}) \leq \|\boldsymbol{X} - \boldsymbol{Y}\|_F \leq \|\boldsymbol{X}\|_F + \|\boldsymbol{Y}\|_F \leq 2 = \frac{2 \cdot c_\phi \cdot \eta}{c_\phi \cdot \eta} \leq \frac{2\cdot}{c_\phi \cdot \eta}\epsilon.$$

Thus, in conclusion, we can assume without loss of generality that equation 22 is satisfied.

Now, returning to equation 21, and using the notation from equation 19, we see that for some permutation $\tau \in S_n$ which fixes the first $d-1$ coordinates, we have that

$$c_\phi^{-1}\epsilon \geq \|h_\mathbf{i}^{(1)}(\boldsymbol{X}) - h_\mathbf{i}^{(1)}(\boldsymbol{Y})\|_\infty = \|\psi \circ F_\mathbf{i}(\boldsymbol{X}) - \psi \circ F_\mathbf{i}(\tau\boldsymbol{Y})\|_\infty \geq c_\psi\|F_\mathbf{i}(\boldsymbol{X}) - F_\mathbf{i}(\tau\boldsymbol{Y})\|_\infty.$$

By also permuting the last $n - d + 1$ coordinates, we may assume without loss of generality that $\tau$ is the identity, so that the bound above give us

$$\|F_{\mathbf{i}}(\boldsymbol{X}) - F_{\mathbf{i}}(\boldsymbol{Y})\|_\infty \leq c_\psi^{-1} c_\phi^{-1} \epsilon, \tag{23}$$

where $\mathbf{i} = (1, \ldots, d - 1)$. For clarity of notation, we henceforth drop the lower index $\mathbf{i}$, and simply write $F$ when referring to $F_{\mathbf{i}}$.

We now want to use equation 23 to bound $d_{\mathcal{G}_+}(\boldsymbol{X}, \boldsymbol{Y})$. For this goal, let $\boldsymbol{R}(\boldsymbol{X})$ be the $SO(d)$ matrix whose first $d - 1$ rows $v_1, \ldots, v_{d-1}$ are obtained from applying the Gram Schmidt procedure to $\boldsymbol{X}$, namely

$$u_1 = x_1$$
$$v_1 = \frac{u_1}{\|u_1\|_2}$$
$$u_2 = x_2 - (x_2 \cdot v_1)v_1$$
$$v_2 = \frac{u_2}{\|u_2\|_2}$$
$$\vdots$$
$$u_{d-1} = x_{d-1} - (x_{d-1} \cdot v_{d-2})v_{d-2} - \ldots - (x_{d-1} \cdot v_1)v_1$$
$$v_{d-1} = \frac{u_{d-1}}{\|u_{d-1}\|_2}$$

and whose last row $v_d$ is defined via

$$u_d = x_{\mathbf{i}}^\perp, \quad v_d = \frac{u_d}{\|u_d\|_2}.$$

Then

$$d_{\mathcal{G}_+}(\boldsymbol{X}, \boldsymbol{Y})^2 \leq \|\boldsymbol{R}(\boldsymbol{X}) \cdot \boldsymbol{X} - \boldsymbol{R}(\boldsymbol{Y}) \cdot \boldsymbol{Y}\|_F^2 = \sum_{j=1}^d \sum_{k=1}^n |v_j(\boldsymbol{X}) \cdot x_k - v_j(\boldsymbol{Y}) \cdot y_k|^2, \tag{24}$$

where $v_j(\mathbf{X})$ denotes the $j$-th' row of $X$, and so now our goal is to bound the right hand side of the inquality above from above by a multiple of $\epsilon$. To do this, we first note that by our assumptions up to now, both $\boldsymbol{X}$ and $\boldsymbol{Y}$ reside in the compact set

$$K = \{\boldsymbol{Y} \in \mathbb{R}^{d \times n} | \frac{1}{2} \leq \|\boldsymbol{Y}\|_F \leq 1 \text{ and } \|y_{\mathbf{i}}^\perp\|_2 \geq \frac{c}{2^d}\}.$$

Our strategy will be to show that, for every $j = 1, \ldots, d$ and $k = 1, \ldots, n$, there exists a smooth function $f_{j,k}$ defined on an open set containing $F(K)$, such that

$$v_j(\boldsymbol{X}) \cdot x_k = f_{j,k}(F(\boldsymbol{X})), \quad \forall \boldsymbol{X} \in K, \tag{25}$$

as a result, we would have that all the $f_{j,k}$ are $L$ Lipschitz on the compact set $F(K)$, for an appropriate $L$, and thus that we can bound the expression on the right hand side of equation 24 by

$$\sum_{j=1}^{d}\sum_{k=1}^{n}|v_j(\boldsymbol{X})\cdot x_k - v_j(\boldsymbol{Y})\cdot y_k|^2 = \sum_{j=1}^{d}\sum_{k=1}^{n}|f_{j,k}(F(\boldsymbol{X})) - f_{j,k}(F(\boldsymbol{Y}))|^2$$

$$\leq \sum_{j=1}^{d}\sum_{k=1}^{n}L^2\|F(\boldsymbol{X}) - F(\boldsymbol{Y})\|_{\infty}^2 \leq L^2 c_{\psi}^{-2}c_{\phi}^{-2}\epsilon^2,$$

which concludes the proof that $H$ is lower Lipschitz on $K_c$. Thus, to conclude step 1(b) of the proof it remains to explain why equation 25 holds for appropriate smooth function $f_{j,k}$.

To do this, let us first show that for $j \in [d-1]$ and $k \in [n]$, the inner product $x_j \cdot x_k$ is a smooth function of $F(\boldsymbol{X})$. Indeed, when $k = i$, we get that $x_j \cdot x_j$ is obtained by squaring $\|x_j\|_2$, which is one of the coordinates of $F(\boldsymbol{X})$. For $k \neq j$, the inner product $x_j \cdot x_k$ is the product of $\frac{x_j \cdot x_k}{\|\boldsymbol{X}\|_F}$, which is a coordinate of $F(\boldsymbol{X})$, and the norm $\|\boldsymbol{X}\|_F$ itself. We have that is a smooth function of $\|x_\ell\|_2$, $\quad \ell = 1, \ldots, n$, which are all coordinates of $F(\boldsymbol{X})$, namely

$$\|\boldsymbol{X}\|_F = \sqrt{\sum_{\ell=1}^{n}\|x_\ell\|_2^2} = N\left(F(\boldsymbol{X})\right)$$

where $N$ denotes the 2-norm function, applied to the appropriate coordinates of $F$. The 2-norm function is smooth on $\mathbb{R}^n \setminus \{0\}$, and the projection of $F(K)$ onto the appropriate $n$ coordinates is contained in $\mathbb{R}^n \setminus \{0\}$ since every $\boldsymbol{X}$ in $K$ has a positive norm. Thus we see that the Frobenius norm, and the inner products $x_j \cdot x_k$, are indeed the composition of a smooth function with $F(\boldsymbol{X})$, where the smooth function is defined in a neighbourhood of $F(K)$.

Now, for $j = 1, \ldots, d-1$, we can show recursively that $v_j(\boldsymbol{X})$ can be written by a linear combination involving smooth functions $\alpha_{s,j}$ applied to the inner products of the first $d-1$ element of $\boldsymbol{X}$, namely

$$v_j(\boldsymbol{X}) = \sum_{s \leq j}\alpha_{s,j}\left((x_a \cdot x_b)_{a,b=1}^{d-1}\right) \cdot x_s.$$

If follows that $v_j(\boldsymbol{X}) \cdot x_k$ is a smooth function of the inner products, which are themselves smooth functions of $F(\boldsymbol{X})$, so we see that equation 25 holds for appropriate smooth $f_{j,k}$.

Finally, we consider the case $j = d$. In this case

$$v_d \cdot x_k = \frac{x_{\mathbf{i}}^{\perp} \cdot x_k}{\|x_{\mathbf{i}}^{\perp}\|_2} = \frac{\|\boldsymbol{X}\|_F}{\|x_{\mathbf{i}}^{\perp}\|_2} \cdot \frac{x_{\mathbf{i}}^{\perp} \cdot x_k}{\|\boldsymbol{X}\|_F}.$$

We note that $\frac{x_{\mathbf{i}}^{\perp} \cdot x_k}{\|\boldsymbol{X}\|_F}$ is a coordinate of $F(\boldsymbol{X})$, while we have already saw that $\|\boldsymbol{X}\|_F$ is a smooth function of $F(\boldsymbol{X})$, and it is known that $\|x_{\mathbf{i}}^{\perp}\|_2$ is non-zero on our domain of interest, and

$$\|x_{\mathbf{i}}^{\perp}\|_2 = \sqrt{\det\left((x_a \cdot x_b)_{a,b=1}^{d-1}\right)},$$

and the inner products themselves are also smooth functions of $F(\boldsymbol{X})$. Thus we have also in the case $j = d$ that equation 25 for an appropriate smooth function $f_{j,k}$. This concludes the proof of Step 1(b).

**Step 2:** We now consider the case where $\|\boldsymbol{X}\|_F = 1$ and $\|\boldsymbol{Y}\|_F \leq 1/2$. We will show that in this case both $d_{\mathcal{G}_+}(\boldsymbol{X}, \boldsymbol{Y})$ and $\|H(\boldsymbol{X}) - H(\boldsymbol{Y})\|_\infty$ are uniformly bounded away from zero and infinity. First, for $d_{\mathcal{G}_+}$, we have for all $\boldsymbol{X}, \boldsymbol{Y}$ which are centralized, that

$$d_{\mathcal{G}_+}(\boldsymbol{X}, \boldsymbol{Y}) \leq \|\boldsymbol{X} - \boldsymbol{Y}\|_F \leq \|\boldsymbol{X}\|_F + \|\boldsymbol{Y}\|_F \leq 3/2$$
$$d_{\mathcal{G}_+}(\boldsymbol{X}, \boldsymbol{Y}) = \min_{\tau \in S_n, \boldsymbol{R} \in SO(d)} \|\boldsymbol{X} - (\tau, \boldsymbol{R})\boldsymbol{Y}\|_F \geq \|\boldsymbol{X}\|_F - \|(\tau, \boldsymbol{R})\boldsymbol{Y}\|_F = \|\boldsymbol{X}\|_F - \|\boldsymbol{Y}\|_F \geq 1/2,$$

where $(\tau_*, R_*)$ above are the group elements minimizing the expression to their left. The upper boundedness of $H$ follows from the fact that all coordinates of $F(\boldsymbol{X})$ are in $[-1, 1]$, and the functions $\psi, \phi$ are Lipschitz, and in particular continuous. Therefore $\|H(\boldsymbol{X})\|_\infty \leq M$ for an appropriate $M$, and so

$$\|H(\boldsymbol{X}) - H(\boldsymbol{Y})\|_\infty \leq 2M.$$

To upper bound $H$ from below, we have

$$1 - 1/4 \leq \|\boldsymbol{X}\|_F^2 - \|\boldsymbol{Y}\|_F^2 = \sum_{j=1}^n \|x_j\|_2^2 - \sum_{j=1}^n \|y_j\|_2^2$$

$$= \sum_{j=1}^n (\|x_j\|_2 - \|y_j\|_2)(\|x_j\|_2 + \|y_j\|_2)$$

$$\leq \frac{3}{2} \sum_{j=1}^n (\|x_j\|_2 - \|y_j\|_2)$$

$$\leq \frac{3n}{2} c_\psi^{-1} c_\phi^{-1} \|H(\boldsymbol{X}) - H(\boldsymbol{Y})\|_\infty$$

so we obtained

$$\|H(\boldsymbol{X}) - H(\boldsymbol{Y})\|_\infty \geq \frac{3}{4} \cdot \frac{2}{3n} = \frac{1}{2n}.$$

We conclude that if $\|\boldsymbol{X}\|_F = 1$ and $\|\boldsymbol{Y}\|_F \leq 1/2$, then

$$\|H(\boldsymbol{X}) - H(\boldsymbol{Y})\|_\infty \geq \frac{1}{2n} \geq \frac{1}{2n} \cdot \frac{2}{3} d_{\mathcal{G}_+}(\boldsymbol{X}, \boldsymbol{Y})$$
$$\|H(\boldsymbol{X}) - H(\boldsymbol{Y})\|_\infty \leq 2M \leq 4M d_{\mathcal{G}_+}(\boldsymbol{X}, \boldsymbol{Y})$$

**Step 3:** To conclude the proof, let $\boldsymbol{X}, \boldsymbol{Y}$ be two non zero point sets in $\Omega_c$. Assume without loss of generality that $\|\boldsymbol{X}\|_F \geq \|\boldsymbol{Y}\|_F$ and set $t = \|\boldsymbol{X}\|_F^{-1}$. Then $t\boldsymbol{X}$ is of norm 1 and $t\boldsymbol{Y}$ is of norm $\leq 1$. If $t\boldsymbol{Y}$ has norm of more than $1/2$ then $t\boldsymbol{X}, t\boldsymbol{Y} \in K_c$, and if not, then $\|t\boldsymbol{X}\|_F = 1, \|t\boldsymbol{Y}\|_F \leq 1/2$. We have upper and lower Lipschitz bounds for both these cases, thus, for an appropriate $0 < c \leq C$ we have that

$$c d_{\mathcal{G}_+}(t\boldsymbol{X}, t\boldsymbol{Y}) \leq \|H(t\boldsymbol{X}) - H(t\boldsymbol{Y})\|_\infty \leq C d_{\mathcal{G}_+}(t\boldsymbol{X}, t\boldsymbol{Y}).$$

Since both $H$ and $d_{\mathcal{G}_+}$ are homogenous, we deduce that the same inequality holds even when we do not scale by $t$, namely

$$c d_{\mathcal{G}_+}(\boldsymbol{X}, \boldsymbol{Y}) \leq \|H(\boldsymbol{X}) - H(\boldsymbol{Y})\|_\infty \leq C d_{\mathcal{G}_+}(\boldsymbol{X}, \boldsymbol{Y}). \tag{26}$$

Finally, it remains to address the case that $\boldsymbol{Y} = 0_{d \times n}$. If also $\boldsymbol{X} = 0_{d \times n}$ then all expressions in equation 26 equal zero and the inequality holds. Otherwise, $\boldsymbol{X}$ is not zero, and then equation 26 holds when replacing $\boldsymbol{Y}$ with $t\boldsymbol{X}$, for all $t > 0$. Taking the limit $t \to 0$ we get the inequality in equation 26 for the case $\boldsymbol{Y} = 0_{d \times n}$, which concludes the proof. □

## 5 Experiments

| Noise Level | EGNN Accuracy | GramNet Accuracy |
|:-:|:-:|:-:|
| 0.0 | 1.0000 | **1.0000** |
| 0.005 | 0.8917 | **0.9390** |
| 0.01 | 0.8546 | **0.8871** |
| 0.05 | 0.5978 | **0.6428** |
| 0.1 | 0.4032 | **0.4953** |

Table 1: Accuracy of EGNN and GramNet under increasing noise levels.

In this section, our goal is to give some indication to the potential of our bi-Lipschitz geometric models. To this end, we evaluate our 1-WL geometric Bi-Lipshcitz model on a task of matching between pairs of point clouds with $d = 2$. In this task, we are given a pair $\boldsymbol{X}, \boldsymbol{Y}$ which are similar, possibly up to rotation and permutation. Namely, $\boldsymbol{X} \approx \boldsymbol{RYP}$ for some rotation matrix $\boldsymbol{R}$ and permutation matrix $\boldsymbol{P}$. The goal is to recover the parameters $(\boldsymbol{R}, \boldsymbol{P})$ and thus the correct matching between $\boldsymbol{X}$ and $\boldsymbol{Y}$ (we handle possibe translation ambiguity by pre-centralizing $\boldsymbol{X}$ and $\boldsymbol{Y}$).

One of the popular approaches to this problem is finding the minimizing group elements from the definition of the PM metric in equation 1. This is a non-convex problem which can be challenging to minimize directly, although closed form solutions are known when minimizing only over permutations, or only over rigid motions. Alternating between these two closed form solutions yields the well known ICP algorithm Besl & McKay (1992) which performs well when initialized properly, but generally can suffer from many local minima.

Due to these challenges, it has been suggested to use machine learning to learn the correct transformations based on the given data (see e.g., Wang & Solomon (2019) ). A common way to perform this is by constructing a parameteric model $f_\theta$ which is invariant to rotations, and equivariant to permutations, in the sense that $f_\theta(\boldsymbol{RYP}) = f_\theta(\boldsymbol{Y})\boldsymbol{P}$. In this case, $f_\theta(\boldsymbol{X})$ and $f_\theta(\boldsymbol{Y})$ will be related only by a permutation, and one can solve for the permutation only using the Sinkhorn algorithm Cuturi (2013) or other approximations of the linear assignment problem. We use this approach here as well (Alternatively, one could also consider permutation invariant and rotation equivariant methods, and solve for the rotation component). Our goal is to compare our proposed bi-Lipschitz equivariant model in the $d = 2$ setting, with a standard equivariant model, EGNN Satorras et al. (2021). Our hypothesis is that since this task is strongly related to the PM metric, the bi-Lipschitzness which our method enjoys will lead to improved performance.

To conduct this experiment, we simulated training and test data, each consisting of $N = 1,000$ pairs of point clouds $(\boldsymbol{X}_k, \boldsymbol{Y}_k) \subset \boldsymbol{R}^2$, where $\boldsymbol{Y}_k$ is a perturbed version of $\boldsymbol{X}_k$. Each such pair was created as follows. First, we generated a 2D Gaussian mixture with three equally weighted factors, where the means of these factors were randomly sampled from a uniform distribution over $[-6, 6]^2$. The covariance matrix of each factor was constructed by generating a 2D diagonal matrix with

diagonal elements sampled from a uniform distribution over $[0, 1]$, and then conjugating it with a random 2D orthogonal matrix. Then, we generated the point cloud $\boldsymbol{X}_k$ by sampling $n = 90$ points from the mixture, and its perturbed version $\boldsymbol{Y}_k$ by independently translating each point in $\boldsymbol{X}_k$ in a random direction. Finally, the point cloud $Y_k$ was rotated by a random angle $\alpha_k$ and permuted by a random permutation $\boldsymbol{P}_k$. In perturbing the point clouds for each new pair $(\boldsymbol{X}_k, \boldsymbol{Y}_k)$, we increased the magnitude of the translations (i.e., the magnitude of the translation vector) linearly in $i$, starting from 0.01 and gradually reaching 0.1. This method, known as "curriculum learning" Bengio et al. (2009), was designed to enable the network to gradually learn the difficult task of matching permuted point clouds in a noisy setting.

Next, we constructed and trained an architecture to match the pairs in the test data, as follows. First, all of the point cloud pairs $(\boldsymbol{X}_k, \boldsymbol{Y}_k)$ were centered. Then, we fed the dataset into our Bi-Lipschitz generalized 1-WL network to generate a permutation-equivariant and rotationally-invariant embedding for each point cloud: in othe words, this network produces rotation invariant node features $h_i(\boldsymbol{X}_k)$ and $h_i(\boldsymbol{Y}_k)$ for each node $i$. The resulting features for each pair were then fed into a differentiable Sinkhorn matching algorithm Cuturi (2013), which outputs a doubly-stochastic matrix $\boldsymbol{Q}_k$ that closely approximates the permutation matrix that best matches the features $h_i(\boldsymbol{X}_k)$ and $h_i(\boldsymbol{Y}_k)$. The training loss measures the quality of $\boldsymbol{Q}_k$ as an approximation for the ground truth permutation $\boldsymbol{P}_k$ which relates $\boldsymbol{X}_k$ to $\boldsymbol{Y}_k$, using a cross entropy loss (applied to each matrix row separately). To use a doubly-stochastic matrix $\boldsymbol{Q}$ outputted by our trained model for matching an out-of-sample pair, we simply compute the permutation $\boldsymbol{P}$ whose each $i$-th row is a one-hot vector such that its '1' entry is at the location of the entry with maximal magnitude in the $i$-th row of $\boldsymbol{Q}$ (in general this procedure produces a matrix with a single 1 in each row, but it may not be a permutation since the same column could have several ones).

The results of both our method and EGNN are shown in Table 1, where we see that our architecture outperforms EGNN in all magnitudes of perturbations. We feel these results indicate the potential of bi-Lipschitz analysis for improving equivariant models, for matching tasks as well as other domains. We believe these empirical results could be improved by constructing equivariant models whose bi-Lipschitz distortion is low. This would require additional tools to those used here, where our focus was mostly on determining bi-Lipschitzness without addressing the constants, and is thus an interesting avenue for future work.

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

## A  From $\mathcal{G}_+$ bi-Lipschitz models to $\mathcal{G}_+$ bi-Lipschitz models

In this appendix, we explain how to transform a $\mathcal{G}_+$ bi-Lipschitz model to a $\mathcal{G}_+$ bi-Lipschitz models, via the following proposition

**Proposition A.1.** *Let $f : \mathbb{R}^{d \times n} \to \mathbb{R}^m$ be a $\mathcal{G}_+$ invariant model, which is bi-Lipschitz with respect to $d_{\mathcal{G}_+}$. Fix $\boldsymbol{R}_0$ to be some rotation matrix with determinant of $-1$ and $\boldsymbol{R}_0^2 = I_d$, and let $\psi : \mathbb{R}^{m \times 2} \to \mathbb{R}^k$ be a $S_2$ invariant and bi-Lipschitz function. Then the function*

$$\tilde{f}(\boldsymbol{X}) = \psi\left(f(\boldsymbol{X}), f(\boldsymbol{R}_0 \boldsymbol{X})\right)$$

*is $\mathcal{G}_\pm$ invariant, and bi-Lipschitz with respect to $d_{\mathcal{G}_\pm}$.*

*Proof.* To see that $\tilde{f}$ is $\mathcal{G}_\pm$ invariant, first note that the permutation and translation invariance follows from the permutation and translation invariance of $f$. For invariance to orthogonal transformation, let $\boldsymbol{R}$ be some orthogonal matrix. If $\boldsymbol{R}$ is also in $SO(d)$, then due to the $SO(d)$ invariance of $f$,

$$\tilde{f}(\boldsymbol{R}\boldsymbol{X}) = (f(\boldsymbol{R}\boldsymbol{X}), f(\boldsymbol{R}_0\boldsymbol{R}\boldsymbol{X})) = (f(\boldsymbol{X}), f([\boldsymbol{R}_0\boldsymbol{R}\boldsymbol{R}_0]\boldsymbol{R}_0\boldsymbol{X})) = \tilde{f}(\boldsymbol{R}\boldsymbol{X})$$

If $\boldsymbol{R}$ is not in $SO(d)$, then $\boldsymbol{R}_0\boldsymbol{R}$ and $\boldsymbol{R}\boldsymbol{R}_0$ are in $SO(d)$, and so

$$\tilde{f}(\boldsymbol{R}\boldsymbol{X}) = \psi\left(f(\boldsymbol{R}\boldsymbol{R}_0)\boldsymbol{R}_0\boldsymbol{X}), f(\boldsymbol{R}_0\boldsymbol{R}\boldsymbol{X})\right) = \psi\left(f(\boldsymbol{R}_0\boldsymbol{X}), f(\boldsymbol{X})\right) = \tilde{f}(\boldsymbol{X}),$$

where for the last equality we used the permutation invariance of $\psi$.

Now, to prove bi-Lipschitzness. By assumption, for appropriate positive $c_f, C_f$, we have that for all $\boldsymbol{X}, \boldsymbol{Y} \in \mathbb{R}^{d \times n}$,

$$c_f d_{\mathcal{G}_+}(\boldsymbol{X}, \boldsymbol{Y}) \le \|f(\boldsymbol{X}) - f(\boldsymbol{Y})\|_2 \le C_f d_{\mathcal{G}_+}(\boldsymbol{X}, \boldsymbol{Y}).$$

We want to use this to prove bi-Lipschitzness of $\tilde{f}$ with respect to $d_{\mathcal{G}_\pm}$. For lower Lipschitzness, we have for all $\boldsymbol{X}, \boldsymbol{Y} \in \mathbb{R}^{d \times n}$

$$
\begin{aligned}
\|\tilde{f}(\boldsymbol{X}) - \tilde{f}(\boldsymbol{Y})\|_2 &= \|\psi\left(f(\boldsymbol{X}), f(\boldsymbol{R}_0\boldsymbol{X})\right) - \psi\left(f(\boldsymbol{Y}), f(\boldsymbol{R}_0\boldsymbol{Y})\right)\|_2 \\
&\ge c_\psi \min\{\|f(\boldsymbol{X}) - f(\boldsymbol{Y})\|_2 + \|f(\boldsymbol{R}_0\boldsymbol{X}) - f(\boldsymbol{R}_0\boldsymbol{Y})\|_2 \\
&\quad, \|f(\boldsymbol{X}) - f(\boldsymbol{R}_0\boldsymbol{Y})\|_2 + \|f(\boldsymbol{R}_0\boldsymbol{X}) - f(\boldsymbol{Y})\|_2\} \\
&\ge c_\psi c_f \min\{2d_{\mathcal{G}_+}(\boldsymbol{X}, \boldsymbol{Y}), 2d_{\mathcal{G}_+}(\boldsymbol{X}, \boldsymbol{R}_0\boldsymbol{Y})\} \\
&= 2c_\psi c_f d_{\mathcal{G}_\pm}(\boldsymbol{X}, \boldsymbol{Y}).
\end{aligned}
$$

To show upper Lipschitzness, we use a similar procedure to obtain

$$
\begin{aligned}
\|\tilde{f}(\boldsymbol{X}) - \tilde{f}(\boldsymbol{Y})\|_2 &= \|\psi\left(f(\boldsymbol{X}), f(\boldsymbol{R}_0\boldsymbol{X})\right) - \psi\left(f(\boldsymbol{Y}), f(\boldsymbol{R}_0\boldsymbol{Y})\right)\|_2 \\
&\le C_\psi \min\{\|f(\boldsymbol{X}) - f(\boldsymbol{Y})\|_2 + \|f(\boldsymbol{R}_0\boldsymbol{X}) - f(\boldsymbol{R}_0\boldsymbol{Y})\|_2 \\
&\quad, \|f(\boldsymbol{X}) - f(\boldsymbol{R}_0\boldsymbol{Y})\|_2 + \|f(\boldsymbol{R}_0\boldsymbol{X}) - f(\boldsymbol{Y})\|_2\} \\
&\le C_\psi C_f \min\{2d_{\mathcal{G}_+}(\boldsymbol{X}, \boldsymbol{Y}), 2d_{\mathcal{G}_+}(\boldsymbol{X}, \boldsymbol{R}_0\boldsymbol{Y})\} \\
&= C_\psi C_f 2 d_{\mathcal{G}_\pm}(\boldsymbol{X}, \boldsymbol{Y}).
\end{aligned}
$$

$\square$

