# OpenReview forum: "Toward bilipshiz geometric models"
_TMLR — Withdrawn by Authors_

### Review · Reviewer_Nkbc · 2025-12-03

**Summary Of Contributions:**

The main takeaway from this article is -- *"One should aim for bi-Lipschitz equivalence when designing models for point clouds"* -- (based on my understanding).

The argument for this proceeds with a sequence of rigorous mathematical statements establishing fundamental theoretical limitations and proposing architectural solutions:

**Contributions:**
- **Theorem 3.1**: The Procrustes Matching (PM) metric and Hard-Gromov-Wasserstein (HGW) distances are *not* bi-Lipschitz equivalent and cannot be. Specifically, they are related by a Hölder exponent of 2 rather than 1, establishing the inequality $d_{\mathcal{D}} \lesssim d_{\mathcal{G}}$ and $d_{\mathcal{G}}^2 \lesssim d_{\mathcal{D}}$.
- **Theorem 4.1**: Standard 1-WL models (graph neural networks) cannot be lower Lipschitz with respect to $d_{\mathcal{G}_{\pm}}$, proving a fundamental impossibility result.
- **Architectural Solution**: To achieve bi-Lipschitz properties w.r.t $d_{\mathcal{G}_{+}}$, they extend models to include well-designed dot products instead of norms using local frames based on $(d-1)$-tuples (initially demonstrated for 2D).
- **Generalization**: Extension of the approach to higher dimensions through systematic construction of local coordinate frames.

**Additional Comments:**

No Additional Comments.

**Audience:**

Yes

**Audience Explanation:**

Efficient design principles for equivariant learning represent a problem of significant current interest in the geometric deep learning. Insights about limitations of MPNN and the relationship between different geometric metrics is a valuable contribution and helps in understanding the properties of invariant neural architectures.

**Broader Impact Concerns:**

None.

**Claims And Evidence:**

No

**Claims Explanation:**

Actually, the answer is both **Yes and No** --

**Theoretical rigor is excellent.** Each contribution is rigorously formulated and, after detailed examination, the proofs appear mathematically sound. The theoretical impossibility results for standard message-passing networks are particularly valuable, providing crucial insights into fundamental limitations of current architectures.

**However, there is insufficient evidence for the main practical takeaway** - "...Aiming for bi-Lipschitz..."

- **Computational complexity analysis**: What is the computational cost of the proposed approach? The method requires processing $(d-1)$-tuples, implying at least $O(n^2)$ complexity for $d=3$ cases. While the authors acknowledge increased computational complexity, the question of "by how much?" remains unaddressed. No runtime or memory comparisons with existing approaches (1-WL, EGNN) are provided.

- **Limited empirical evidence**: The claimed advantages of bi-Lipschitz properties are demonstrated only through a minimal synthetic toy example (Table 1). There is no substantial evidence that bi-Lipschitz properties provide significant advantages in realistic scenarios through either:
  - (a) Additional mathematical statements or proofs establishing practical benefits, or
  - (b) Comprehensive empirical evaluation on real-world datasets

- **Inadequate baselines**: Only comparison with EGNN is provided. Missing comparisons with other frame-based methods (Vector Neurons, Frame Averaging, Tensor Field Networks) that may also achieve similar stability properties.

- **Limited dataset scope**: Evaluation restricted to synthetic Gaussian mixture matching tasks. Missing standard benchmarks.

- **Missing validation of core claims**: No direct empirical measurement of the bi-Lipschitz property (maybe correlation plots between input PM distances and embedding distances) to validate the theoretical claims.

**Requested Changes:**

The following are only suggestions. Not all of them are required. Please choose the ones suitable to your requirements.

1. Comprehensive computational complexity analysis -- Provide detailed runtime and memory complexity analysis comparing the proposed GramNet with existing approaches (1-WL, EGNN). And maybe add a paragraph discussing practical implications of the complexity accuracy tradeoff for real-world applications

2. empirical validation -- Can we empirically validate theorem 3.1?

3. better experimental design? -- Ablation studies validating $\phi, \psi$ functions and what happens if the assumptions (used for theoretical results) for these do not hold. Validation for 3d point clouds as well.

4. Some typos I found which must be corrected. Preferably a more thorough checking.
- "bilipshiz" → "bilipshitz" in several places
- "natural n,d" → "natural numbers n,d"
- Wrong reference in the proof of Theorem 4.1
- Inconsistent notation : $d\_{\\mathcal{G}\_{\\pm}}$  or  $d\_{\\mathcal{G}\_{+}}$ in the appendix and Theorem 4.1

---

### Review · Reviewer_WQ2L · 2025-12-13

**Summary Of Contributions:**

The paper proposes a bi-Lipschitz message passing neural network (MPNN) operating on point clouds. A bi-Lipschitz function is defined as a mapping with both lower and upper Lipschitz bounds. The authors additionally require the model to be invariant with respect to the group $G_\pm = \mathcal{O}(d) \rtimes \mathbb{R}^d \times S_n$.  They analyze the $\mathcal{G}_\pm$​-invariance properties of the Procrustes Matching (PM) and Hard Gromov–Wasserstein (HGW) metrics, and establish a bi-Hölder relationship between them. The PM metric is bi-Hölder with respect to the HGW metric, but the first Hölder exponent must be larger than $1$, which implies that the two metrics cannot be bi-Lipschitz. The authors motivate the necessity of a new model by highlighting limitations of existing MPNN architectures. For this, they formally show that existing MPNNs do not have a lower bound w.r.t. PM metric. The new architecture is first introduced for graphs in a two-dimensional setting, and subsequently for general $d$-dimensional settings. A single experiment on a synthetic dataset is conducted, which illustrates one scenario in which the suggested architecture performs better than EGNNs [1].

- **Strengths:**
	- S1: The definition of the group via $G_\pm = \mathcal{O}(d) \rtimes \mathbb{R}^d \times S_n$ allows the reader to understand the intuition behind what transformations the model shall become invariant against.
	- S2: Section "1.3 Definitions" is well structured.
	- S3: The selected topic addresses current problems for MPNNs.
- **Weaknesses:**
	- W1: The literature research presented in Section 1.2 "Related work" omitts a lot of relevant work about the subjects of group-in- and -equivariance of ML models such as [8, 9], work on isometry invariance (which include rotations and translations) such as surface learning approaches [10, 12], fundamental work about permutation invariant processing of point-cloud data such as [6, 7, 11] and message passing neural networks such as [5]. Instead, the related work section is very brief and references are scattered throughout later sections.
	- W2: A meaningful discussion of $G_\pm$-invariant models requires (A) an explicit definition of a left or right group action of $\mathcal{G}_\pm$ and (B) a formal proof that the model-defined operations are invariant under this action. Neither is provided in sufficient detail.
	- W3: The choice of the group $G_\pm$ is insufficiently motivated. Prior work, such as [4], motivates the necessity of bi-Lipschitz models with the phenomenon of over-squashing in long-range tasks. Given such a bi-Lipschitz model, it remains unclear what additional benefits are obtained by enforcing $\mathcal{G}_\pm$-invariance on top of bi-Lipschitzness.
	- W4: The detailed discussion about the bi-Hölder and bi-Lipschitzness relationship between the PM and HGW metrics (Section 2.1) appears to be disconnected from the ultimate goal of constructing a bi-Lipschitz, $\mathcal{G}_\pm$-invariant MPNN.
	- W5: Given that the goal of [4] is also to design bi-Lipschitz MPNNs, a conceptual proximity is given. Therefore, a direct comparison, both theoretically and experimentally, is well motivated but missing.
	- W6: The manuscripts structure generally appears incoherent. Some of the reasons are:
		- Several digressions interrupt the logical flow, such as the discussion of centralization in between the introductions of the PM and HGW metrics in Section 2
		- Frequent digressions into lengthy proofs spanning over multiple pages
		- A variety of typographical and grammatical errors, including errors in the title of the manuscript
	- W7: The experimental evaluation is extremely limited. Only a single experiment on a synthetic dataset, that contains point clouds with just 90 elements, is conducted. This is insufficient to validate the proposed architecture.
	- W8: No code has been provided that allows for the reproduction of reported experiments.
	- W9: Theorem 3.1 shows that MPNNs have an upper Lipschitz bound w.r.t. the HGW metric and Corollary 3.2 suggests that MPNNs cannot have a lower Lipschitz bound w.r.t. the PM metric. According to Definition 1.3, this does not constitute evidence that common MPNNs are non bi-Lipschitz w.r.t. a single metric, since different metrics have been used for the lower and upper bounds.

**Additional Comments:**

**References:**

[1] Satorras, Víctor Garcia, Emiel Hoogeboom, and Max Welling. “E(n) Equivariant Graph Neural Networks.” _International Conference on Machine Learning_. PMLR, 2021.

[2] Bogo, Federica, et al. "FAUST: Dataset and evaluation for 3D mesh registration." _Proceedings of the IEEE conference on computer vision and pattern recognition_. 2014.

[3] Verine, Alexandre, et al. "On the expressivity of bi-Lipschitz normalizing flows." _Asian Conference on Machine Learning_. PMLR, 2023.

[4]  Sverdlov, Yonatan, et al. ‘FSW-GNN: A Bi-Lipschitz WL-Equivalent Graph Neural Network’. _The Fourth Learning on Graphs Conference_, 2025, openreview.net/forum?id=zVXZK5JRHB.

[5] Kipf, Thomas N., and Max Welling. ‘Semi-Supervised Classification with Graph Convolutional Networks’. _International Conference on Learning Representations_, 2017, openreview.net/forum?id=SJU4ayYgl.

[6] Qi, Charles R., et al. "Pointnet: Deep learning on point sets for 3d classification and segmentation." _Proceedings of the IEEE conference on computer vision and pattern recognition_. 2017.

[7] Qi, Charles Ruizhongtai, et al. "Pointnet++: Deep hierarchical feature learning on point sets in a metric space." _Advances in neural information processing systems_ 30 (2017).

[8] Cohen, Taco S., and Max Welling. "Steerable cnns." _arXiv preprint arXiv:1612.08498_ (2016).

[9] Cohen, Taco, and Max Welling. "Group equivariant convolutional networks." _International conference on machine learning_. PMLR, 2016.

[10] Haan, Pim De, et al. ‘Gauge Equivariant Mesh CNNs: Anisotropic Convolutions on Geometric Graphs’. _International Conference on Learning Representations_, 2021, openreview.net/forum?id=Jnspzp-oIZE.

[11] Zaheer, Manzil, et al. ‘Deep Sets’. _Advances in Neural Information Processing Systems_, edited by I. Guyon et al., vol. 30, Curran Associates, Inc., 2017, proceedings.neurips.cc/paper_files/paper/2017/file/f22e4747da1aa27e363d86d40ff442fe-Paper.pdf.

[12] Masci, Jonathan, et al. ‘Geodesic Convolutional Neural Networks on Riemannian Manifolds’. _Proceedings of the IEEE International Conference on Computer Vision (ICCV) Workshops_, 2015.

[13] Morris, Christopher, et al. "Position: Future directions in the theory of graph machine learning." _Forty-first International Conference on Machine Learning_. 2024.

**Audience:**

No

**Audience Explanation:**

The manuscript addresses a variety of topics that have been studied in prior work: the subject of in- and equivariances has been studied by [8] and [9]. Invariance towards isometries acting on 3D shapes has been addressed by work that deals with learning surface data such as [10] or [12]. Permutation-invariant processing of point-clouds has been addressed by [6],  [7] and [11]. Fundamental theory about graph- or message passing neural networks, respectively, has been described in [5]. Also, as indicated by the position paper [13], there exists a growing interest in bi-Lipschitz MPNNs, which is also indicated by new publications such as [3] or [4]. Therefore, a interest in the subject studied by the manuscript is well justified.

However, the paper lacks a comprehensive literature review that clearly positions the proposed approach relative to existing work. For example, [4] introduces a bi-Lipschitz MPNN, yet the manuscript does not address the conceptual differences to the proposed methodology. Without a comparison and a clear articulation of the novelty, the benefits of the findings presentend in this paper remain unclear.

**Broader Impact Concerns:**

None.

**Claims And Evidence:**

No

**Claims Explanation:**

According to Section 1.1 "Main Results", the paper focuses on showing:
1. **The PM metric and the HGW metric are not bi-Lipschitz.**
	- Theoretical evidence:
		- Theorem 2.1 shows that the PM metric is bi-Hölder w.r.t. the HGW metric.
		- Theorem 2.4 shows that the first Hölder exponent has to be larger then $1$. Thus, the PM and HGW metrics cannot be bi-Lipschitz.
2. **Common MPNNs are not bi-Lipschitz w.r.t. the PM metric.**
	- Theoretical evidence:
		- Theorem 3.1 shows that MPNNs have an upper Lipschitz bound w.r.t the HGW metric.
		- Corollary 3.2 suggests that MPNNs cannot have a lower Lipschitz bound w.r.t. the PM metric.
		- See W9 for the comment on why these findings are insufficient to show non-bi-Lipschitzness of existing MPNNs.
3. **A modification of MPNNs to become bi-Lipschitz.**
	- Theoretical evidence:
		- In Section 4, the authors introduce their architecture for the planar case. Bi-Lipschitzness is established by showing that the message-passing function $\psi$ and the readout function $\phi$ are bi-LIpschitz w.r.t. $p$-norms and $p$-Wasserstein distances.
		- In Section 4.1, the authors extend their model for the $d$-dimensional case as a natural extension of their planar approach and show via Theorem 4.2 that its bi-Lipschitz.

**Practical evidence:**
- From a practical point of view, only a single experiment on a synthetic dataset has been conducted, which is summerized in Table 1.
- Table 1 contains a noise column. The manuscript does not explain what the noise has been applied to.
- Standard correspondence benchmarks, such as the FAUST-benchmark [2], are not considered.
- The dataset contains point clouds with only $90$ points, which is a lot less compared to point cloud data in common benchmarks.
- Experiments are restricted to the planar case, that is $d=2$, despite the claim of applicability to general $d$-dimensional settings.
- No code has been provided. Thus, reproducing the experiment results is not possible.
- The proposed architecture has only been compared against the EGNN model [1].

**Requested Changes:**

Given the weaknesses described below the paper summary, the paper exhibits a variety of aspects that need to be changed:
1. W1: Please extend the related work section, such that it sufficiently addresses the background subjects that are related to the topic addressed by this manuscript.
2. W2: Please provide a definition of a left- or right group action for the Group $\mathcal{G}_\pm$ and a formal proof which shows that the model developed in Section 4 is in fact invariant towards the defined group actions.
3. W3: Please motivate the necessity of making bi-Lipschitz MPNNs additionally invariant towards actions of $\mathcal{G}_\pm$.
4. W4: Please provide a discussion that explains the link between showing that the PM and HGW metrics are not bi-Lipschitz (Section 2.1) and the suggested MPNN (section 4).
5. W5: Please add a discussion and experiments that highlight the differences between the bi-Lipschitz model suggested in this manuscript and the bi-Lipschitz MPNN described in [4].
6. W6: Please improve the structure of the manuscript and its readability by:
	1. correcting all typographical and grammar mistakes
	2. moving multi-page proofs into the appendix, only keeping the theorems in the main body of the manuscript
	3. moving digressions, such as the discussion of centralizing point clouds in between of introducing two metrics, before or after the main point of discussion
7. W7: Please conduct more experiments to validate the suggested model:
	1. conduct experiments on standard benchmarks datasets for correspondence tasks (e.g., FAUST benchmark [2])
	2. comparing to more models, in particular to the bi-Lipschitz model proposed by [4]
	3. validate the suggested model for a scenario in which $d \geq 3$.
8. W8: Please provide code that allows to reproduce the reported results of the conducted experiments.

---

### Review · Reviewer_LQiS · 2025-12-15

**Summary Of Contributions:**

The paper investigates the bi-Lipschitz property of invariant networks for point cloud data. The authors demonstrated that the message passing architecture is in general not bi-Lipschitz, thus they do not guarantee the quality of separation in the input space. They shown how to obtain bi-Lipschitz models in the d-dimensional case.

**Audience:**

Yes

**Audience Explanation:**

The topic is of interest, especially for inverse problems

**Claims And Evidence:**

Yes

**Claims Explanation:**

The theorems of the first three sections are convincing. However, for section 4, the theorems refer to a specific model, whose properties should be investigated also numerically.

**Requested Changes:**

Major points that should be addressed:
1) The experiments section is weak. I suggest to investigate your setting for different n, and include at least d=3. Would also be nice to report computational time of the two methods as a function of n, d.
2) On the computational complexity: eq. (11-13) seems to have quadratic cost in n. For the d-dimensional case, the cardinality of T seems to be the binomial coefficient of (n, d). Comment on this and implication for the method's applicability.

Minor
0) "bilipshiz" in the title. Is this a typo?
1) I suggest to be a bit more clear in the motivations of the paper and the importance of chasing the bi-bilipshiz property. You start to explain completeness but it's not needed to understand the importance of preserving distances in a group transformation preserving network; the motivation of your work is introduced in the manuscript at "However, completeness guarantees do not address the quality of separation". I'd state this earlier; maybe an illustrative example could help the reader.
2) It is not clear if a (easy) neural parametrization (as alternative to the beta function) is still possible for multiset-to-vector functions which are both bi-Lipschitz and homogenous. Please comment on this.
3) Equation (8): previously, $\psi$ was a function applied to a multiset, taking inputs $c_i$, $c_j$, $\Vert x_i-x_j\Vert_2$. In equation (8), accepts input $\Vert x_i\Vert_2$, and a different multiset. It's a bit confusing. Where is the node feature? Was $x_j$ supposed to be $c_j$ here?
4) Before eq (10): Shouldn't it be $\phi(t S)=t \phi(S)$?
5) You could stress, soon after introducing it, that (9) is now a function taking as input a multiset, as opposed to 1-WL geometric model, which requires the p-Wasserstein bi-Lipschitz condition.

Very minor
"Accordingly, an invariant model" If it is a main result, should be phrased as "Accordingly, we show that...", if it is following from your previous statement instead, is not clear why it should be obvious.
"is needs" it needs
"are concludes" are concluded

---

### Note · Authors · 2026-02-01

**Comment:**

we would like to thank the reviewers and area chair for the time taken to review our manuscript. We have decided to withdraw that paper and send it to a more mathematical venue.

**Withdrawal Confirmation:**

I have read and agree with the venue's withdrawal policy on behalf of myself and my co-authors.